# The Risk for Neonatal Hypoglycemia and Bradycardia after Beta-Blocker Use during Pregnancy or Lactation: A Systematic Review and Meta-Analysis

**DOI:** 10.3390/ijerph19159616

**Published:** 2022-08-04

**Authors:** Rosalie de Bruin, Sarah L. van Dalen, Shamaya J. Franx, Viraraghavan V. Ramaswamy, Sinno H. P. Simons, Robert B. Flint, Gerbrich E. van den Bosch

**Affiliations:** 1Faculty of Medicine, Erasmus University Medical Center, 3015GD Rotterdam, The Netherlands; 2Department of Neonatology, Ankura Hospital for Women and Children, Hyderabad 500072, Telangana, India; 3Department of Pediatrics, Division of Neonatology, Erasmus University Medical Center, 3015GD Rotterdam, The Netherlands; 4Department of Hospital Pharmacy, Erasmus University Medical Center, 3015GD Rotterdam, The Netherlands

**Keywords:** beta-blockers, bradycardia, hypoglycemia, lactation, neonate, pregnancy

## Abstract

Beta-blockers are often used during pregnancy to treat cardiovascular diseases. The described neonatal side effects of maternal beta-blocker use are hypoglycemia and bradycardia, but the evidence base for these is yet to be evaluated comprehensively. Hence, this systematic review and meta-analysis was performed to evaluate the potential increased risk for hypoglycemia and bradycardia in neonates exposed to beta-blockers in utero or during lactation. A systematic search of English-language human studies was conducted until 21 April 2021. Both observational studies and randomized controlled trials investigating hypoglycemia and/or bradycardia in neonates following beta-blocker exposure during pregnancy and lactation were included. All articles were screened by two authors independently and eligible studies were included. Pair-wise and proportion-based meta-analysis was conducted and the certainty of evidence (CoE) was performed by standard methodologies. Of the 1.043 screened articles, 55 were included in this systematic review. Our meta-analysis showed a probable risk of hypoglycemia (CoE—Moderate) and possible risk of bradycardia (CoE—Low) in neonates upon fetal beta-blocker exposure. Therefore, we suggest the monitoring of glucose levels in exposed neonates until 24 h after birth. Due to the limited clinical implication, monitoring of the heart rate could be considered for 24 h. We call for future studies to substantiate our findings.

## 1. Introduction

Beta-blockers are often used antihypertensive agents during pregnancy to treat pre-existing hypertension, pregnancy induced hypertension, pre-eclampsia and tachyarrhythmia. Labetalol and metoprolol are the most prescribed beta-blockers during pregnancy and lactation [1]. As most beta-blockers are known to cross the placenta, their in-utero exposure may affect the condition of the fetus during pregnancy or of the neonate after birth [2,3]. Previous studies have shown that the maternal use of beta-blockers is not associated with a large increase in the risk for either overall malformations or cardiac malformations for the neonate [4,5]. However, it is unknown whether exposure to beta-blockers in utero considerably increases the risk for neonatal side effects such as bradycardia or hypoglycemia. Moreover, some beta-blockers have been found in breast milk [6], which may lead to potential negative effects in the neonate as well. However, the degree of excretion of different beta-blockers into breast milk is dependent on their individual pharmacokinetic characteristics [7]. Beta-blockers such as labetalol and propranolol pass into breastmilk in low concentrations [8,9], while atenolol reaches a relatively higher breastmilk concentration [10]. The question arises as to whether beta-blocker exposure is safe for the neonate.

### 1.1. Mechanisms of Action and Potential Risks

Beta-blockers inhibit the beta-1 and beta-2 receptors resulting in a decrease in heart rate and heart contractility causing a lower blood pressure [11]. Maintaining heart rate within the normal range is vital during the neonatal period as their myocardium has a limited ability to compensate in response to bradycardia by virtue of increasing the stroke volume. Therefore, beta-blocker exposure in utero and through lactation could potentially harm the neonate by decreasing heart rate, which hypothetically could cause decreased cardiac output, blood pressure and eventually organ perfusion and growth. Moreover, beta-blockers cause inhibition of the glycogenolysis through the activation of the sympathetic nervous system [11], which could lead to hypoglycemia. In neonates, hypoglycemia is potentially dangerous, since severe or prolonged hypoglycemia can result in significant insult to the developing brain [12,13].

Therefore, it is important to evaluate the risks associated with exposure to beta-blockers in utero or through lactation for the outcomes of bradycardia and hypoglycemia in neonates. In the case of a distinctly increased risk for the neonate, glucose and/or heartrate monitoring are needed and may require prolonged hospital admission in order to detect and treat these potentially harmful side effects. On the other hand, if the risk for hypoglycemia or bradycardia is not increased, the neonate may not require admission for monitoring, thus mitigating the need for a longer hospital stay and exposure to painful blood collection procedures for glucose tests.

### 1.2. Aim of the Study

The aim of this systematic review and meta-analysis was to evaluate the risk of hypoglycemia and bradycardia in neonates exposed to beta-blockers in-utero or through lactation in comparison with neonates without any beta-blocker exposure in order to assess the need of postnatal observation including heartrate and glucose monitoring. With this systematic review clinicians can make a better risk-benefit judgement for the use of beta-blockers during pregnancy and lactation.

## 2. Materials and Methods

### 2.1. Protocol and Registration

The methods of this systematic review and meta-analysis were specified in our protocol (submitted after minor revision) and registered in PROSPERO (CRD42021264269) on 27 July 2021. The PRISMA 2009 [14] reporting checklist was used.

### 2.2. Patient and Public Involvement

There was no patient or public involvement in the whole process of conducting this research.

### 2.3. Eligibility Criteria

The following inclusion criteria were applied on the articles (including abstracts) resulting from our search:Randomized controlled trials, case series, case reports and observational studies reporting the adverse effects of beta-blockers exposure during pregnancy and lactation on the neonate;The described effect on the neonate should focus on hypoglycemia and/or bradycardia (the chosen cut-off value for hypoglycemia and bradycardia could vary between studies);Studies in English;Studies in human subjects.

Articles were excluded when they met the following criteria:5.Articles studying only the effect of the disease of the mother on the neonate;6.Studies on the effect of beta-blockers on only the fetus and not the neonate;7.Reviews and editorials;8.Letters to the Editor.

### 2.4. Information Sources

Electronic database searches were used to identify studies for this systematic review. A search strategy was developed in consultation with a professional librarian of the Erasmus Medical Center, using the following electronic databases: EMBASE, Medline, Cochrane Central Register of Trials and Web of Science. English-language human studies published since the inception of the databases until 21 April 2021 were included. See the Appendix A for the search strategy and hits.

### 2.5. Search Strategy

Search terms on the following subjects were included (1) beta-blockers, (2) pregnancy or lactation and (3) the effect on the neonate in terms of hypoglycemia and bradycardia. Search terms in MEDLINE included, for instance, ‘beta adrenergic receptor blocking agent/exp’, ‘hypoglycaemia/de’, ‘bradycardia/exp’, ‘perinatal drug exposure’ and ‘lactation/de’ (see Appendix A for the complete search strategies).

### 2.6. Study Selection

Two members of the study team independently reviewed the articles based on title and abstract using the above-described inclusion and exclusion criteria (RB and SF). After a selection based on the title and abstract, two reviewers independently read the full text of the selected articles (RB and SD). Any identified discrepancies between the reviewers were resolved through evaluation and discussion by GB and RF when necessary. Extracted information included the number of participants, participant demographics, study population, study design, outcome (hypoglycemia and/or bradycardia), treatment indication and type and dosage of the beta-blocker. The number of articles meeting the inclusion criteria was recorded and the reasons for exclusion were documented in accordance with the PRISMA guidelines [14].

### 2.7. Data Collection Process

Two authors (RB and SD) extracted the data from included studies and three other authors performed a check on all extracted data (SF, GB and VR). Disagreements were resolved by discussion between the review authors. If no agreement could be reached, the senior researchers decided (GB, RF).

### 2.8. Data Items

Data were extracted from each included study on (1) the characteristics of trial participants (including treatment indication, participant demographics), (2) the type and dosage of the beta-blocker (beta-blocker versus placebo or versus another antihypertensive agent or beta-blocker versus beta-blocker), and (3) the outcome measure (including hypoglycemia and/or bradycardia).

### 2.9. Risk of Bias Assessment

The Cochrane risk of bias tool version 2.0 was used for randomized controlled trials (RCTs) [15] and Risk Of Bias in Non-randomized Studies of intervention (ROBINS-I) for non-RCTs [16]. Two authors (VR, GB) assessed the risk of bias independently and disagreements were resolved by consulting a third author (RF). The quality of case reports was evaluated using the Checklist for Case Reports by the Joanna Briggs Institute (JBI) [17]. The quality of case series was evaluated using the Checklist for Case Series by the JBI [18]. Two authors (RB, RF) assessed the risk of bias for case reports and case series independently and disagreements were resolved by consulting a third author (GB).

### 2.10. Certainty of Evidence Assessment

Grading of Recommendations, Assessment, Development and Evaluations (GRADE) were used for the Certainty of Evidence (CoE) [19,20,21]. CoE was classified into four categories, namely high, moderate, low and very low. The findings of the systematic review are reported as per a modified GRADE working group recommendation [22] (Table 1).

### 2.11. Data Synthesis

R software (version 3.6.2, R Foundation for Statistical Computing, Vienna, Austria) was utilized for data analysis [23]. Meta-analysis was performed by the Mantel Haenszel method and the inverse variance method for dichotomous outcomes and continuous outcomes, respectively. Heterogeneity was assessed using Cochran Q, I^2^ and τ^2^ values. A random effects model was utilized if the I^2^ value was >50%, and if it was ascertained that the large I^2^ values were not due to differences between small and large magnitude of the effect estimates. Else, a fixed effect model was utilized. A random effects meta-analysis of proportions with the Freeman–Tukey Double arcsine transformation was also used.

## 3. Results

### 3.1. Study Selection

The PRISMA flow is shown in Figure 1, starting with 1462 identified articles from our search in EMBASE, Medline, Cochrane Central Register of Trials and Web of Science. After removal of duplicates, 1043 articles remained. Screening the titles and abstracts using the inclusion and exclusion criteria led to excluding 911 articles. Subsequently, 132 articles were screened based on the entire content of the article. The reviewers (RB, SD, SF) had discrepancies about four articles [24,25,26,27] and these were resolved through evaluation and discussion by GB and RF. Two of these [26,27] articles were included. The two excluded articles [24,25] did not report sufficiently on the occurrence of hypoglycemia and bradycardia in neonates exposed to beta-blockers in utero or through lactation. Two articles were not electronically available and were therefore excluded as well. This led to a total of 776 articles being excluded and 55 articles being included in this systematic review, of which 15 are case reports. Table 2 shows the details of 40 of the included studies. The 15 case reports are described in Section 3.6.

All included studies provided information about possible neonatal effects after in utero exposure or through lactation. Few studies provided information about the neonatal outcome after beta-blocker use during lactation; see paragraph 3.5. Moreover, Table 3 shows the type and dosage of the used beta-blocker in each study. The case reports were not included in the meta-analysis. Furthermore, two other studies were excluded from the meta-analysis since these focused on the short-term use of beta-blockers during pregnancy for the induction of anesthesia or labor instead of its use for cardiovascular diseases [28,29].

**Table 2 ijerph-19-09616-t002:** Overview of 40 included studies.

Source	Study Group	Number of Patients	Country	Study Design	Objective	Treatment Indication	Outcome
**Bigelow CA, 2021** [29]	Nulliparous patients undergoing term induction of labor with a single, non-anomalous gestation received propranolol or placebo	*n* = 121 cases vs. *n* = 119 controls	United States	RCT	To determine whether the addition of a single dose of propranolol to induce labor in nulliparous women would decrease total time to vaginal delivery	Induction of labor	Hypoglycemia
**Kayser A, 2020** [30]	Neonates of hypertensive women treated with metoprolol and/or bisoprolol after the first trimester, but not with methyldopa at any time during pregnancy	*n* = 294 cases vs. *n* = 225 controls (methyldopa) vs. *n* = 588 controls (nonhypertensive mothers)	Germany	Cohort study	To evaluate the effects of beta blockers during the second and third trimester on fetal growth, length of gestation and postnatal symptoms in exposed infants	Chronic or pregnancy-induced hypertension	Hypoglycemia and bradycardia
**Kumar N, 2020** [31]	All the infants born ≥34 weeks with mothers using beta blockers prenatally compared to mothers with diabetes, both beta-blockers and diabetes or without pregnancy conditions	*n* = 228 cases (BB) and *n* = 60 both vs. *n* = 379 controls (diabetes), and *n* = 4.103 controls (no pregnancy condition)	United States	Cohort study	To evaluate whether pregnancy glycated hemoglobin (HbA1c) levels of ≤6% and maternal race impacts neonatal hypoglycemia and birthweight, and whether diabetes and beta blocker use during pregnancy additively impacts neonatal outcomes	Not described in article	Hypoglycemia
**Mazkereth R, 2019** [32]	Infants born to mothers who were treated with beta-blockers during pregnancy and until delivery	*n* = 153 cases vs. *n* = 153 controls	Israel	Case–control	To evaluate infants exposed to intrauterine beta blockers in order to estimate the need of postnatal monitoring	Cardiac disease (arrhythmia, rheumatic heart disease and cardiomyopathy), chronic hypertension, migraine, PIH (pregnancy induced hypertension) and pre-eclampsia	Hypoglycemia and bradycardia
**Easterling T, 2019** [33]	Pregnant woman older than 18 years and gestational age of at least 28 weeks received labetalol, nifedipine or methyldopa	*n* = 295 cases vs. *n* = 298 controls (nifedipine) vs. *n* = 301 controls (methyldopa)	India	RCT	To compare the efficacy and safety of oral labetalol, nifedipine retard and methyldopa for the management of severe hypertension in pregnancy	Hypertension in pregnancy	Hypoglycemia and bradycardia
**Thewissen L, 2017** [34]	Preterm neonates prenatally exposed to labetalol because of maternal HDP	*n* = 22 cases vs. *n* = 22 controls with maternal HDP without labetalol and *n* = 22 controls without maternal HDP	Belgium	Case–control	To investigate labetalol-induced effects on neonatal hemodynamics and cerebral oxygenation in the first 24 h after birth	Hypertensive disorders in pregnancy (HDP)	Bradycardia
**Bateman BT, 2016** [35]	Completed pregnancies linked to liveborn infants	*n* = 2,292,116	United States	Cohort study	To define the risks of neonatal hypoglycemia and bradycardia associated with maternal exposure to beta blockers at the time of delivery	Pre-existing or gestational hypertension, pre-eclampsia, migraine, cardiac arrhythmia, ischemic heart disease, anxiety and congestive heart failure	Hypoglycemia and bradycardia
**Singh R, 2016** [36]	Women with severe hypertension in pregnancy who received labetalol or hydralazine	*n* = 50 cases vs. *n* = 50 controls	India	RCT	To evaluate the efficacy and safety of intravenous labetalol and intravenous hydralazine in managing hypertensive emergency in pregnancy	Hypertension in pregnancy	Hypoglycemia
**Heida KY,2012** [37]	Infants from mothers suffering from severe preeclampsia and/or HELLP treated with labetalol	*n* = 55 cases and *n* = 54 controls	The Netherlands	Case–control	Analysis of possible association between intrauterine labetalol exposure and side effects	Preeclampsia and/or HELLP-syndrome	Hypoglycemia and bradycardia
**Verma R, 2012** [38]	Pregnant patients newly diagnosed with systolic blood pressure of ≥140 mmHg and a diastolic blood pressure of ≥90 mmHg and gestational age between 20–40 weeks of pregnancy received labetalol or methyldopa	*n* = 45 cases vs. *n* = 45 controls	India	RCT	(1) To evaluate the effect of labetalol versus methyldopa on maternal outcomes in the treatment of new onset hypertension during pregnancy (2) To evaluate the effect of labetalol versus methyldopa on fetal and neonatal outcomes in the treatment of new onset hypertension during pregnancy	Pregnancy induced hypertension	Hypoglycemia and bradycardia
**Davis RL, 2011** [39]	Women older than 15 years delivering an infant, who had been continuously enrolled with prescription drug coverage for ≥1 year prior to delivery	*n* = 584 cases (full-term infants exposed to beta blockers), *n* = 804 controls (full-term infants exposed to calcium-channel blockers) and >75,000 unexposed infants	United States	Cohort study	To study risks for perinatal complications and congenital defects among infants exposed to beta blockers in utero	Not described in article	Hypoglycemia
**Vigil-De Gracia P,** **2006** [40]	Women with severe hypertension in pregnancy treated with labetalol or hydralazine	*n* = 100 cases vs. *n* = 100 controls (and *n* = 103 case children vs. *n* = 102 control children)	Panama	RCT	To compare the safety and efficacy of intravenous labetalol and intravenous hydralazine for acutely lowering blood pressure in pregnancy	Severe preeclampsia, gestational hypertension, superimposed preeclampsia, chronic hypertension, eclampsia and severe preeclampsia with HELLP	Hypoglycemia and bradycardia
**Darcie S, 2004** [41]	Newborns of mothers treated with atenolol, isradipine or a low sodium diet during pregnancy	*n* = 40 cases vs. *n* = 39 controls (isradipine) vs. *n* = 14 controls (low sodium diet)	Brazil	RCT	To evaluate the effect of isradipine on the evolution of glycemia levels in newborns of pregnant women who have arterial hypertension, comparing it to the use of atenolol and situations where the blood pressure control was done without using antihypertensive medications	Specific hypertensive disease of pregnancy (SHDP) or chronic arterial hypertension and superimposed SHDP	Hypoglycemia
**Paran E, 1995** [42]	Woman with moderate pregnancy-induced hypertension	*n* = 17 cases propranolol/hydralazine vs. *n* = 19 cases pindolol/hydralazine vs. *n* = 13 controls with hydralazine	Turkey	RCT	To compare the effect of propranolol/hydralazine to pindolol/hydralazine combination therapy with hydralazine monotherapy and to evaluate the clinical effects on the mother and on the fetus	Moderate pregnancy-induced hypertension	Hypoglycemia
**Munshi UK, 1992** [43]	Neonates born to mothers suffering from pregnancy-induced hypertension (PIH) and receiving labetalol compared to children of mothers treated with drugs other than labetalol for their PIH	*n* = 48 cases vs. *n* = 81 controls	India	Case–control	To assess the incidence of birth asphyxia, intrauterine growth retardation and hypoglycemia in the neonates of mothers suffering from pregnancy induced hypertension treated with labetalol	Pregnancy induced hypertension	Hypoglycemia and bradycardia
**Bott-Kanner G, 1992** [44]	Women presenting with a diastolic blood pressure of 85–90 mmHg before the 35th week of pregnancy treated with pindolol or placebo	*n* = 30 cases vs. *n* = 30 controls	Israel	RCT	To investigate the benefits of early treatment of hypertension of pregnancy with pindolol and to compare the effects of initiating treatment at a DBP of 85–99 mmHg as opposed to starting treatment when DBP is ≥100 mmHg. The study examined the effects of treatment in incidence of maternal and fetal complications.	A diastolic blood pressure of 85–90 mmHg before the 35th week of pregnancy	Hypoglycemia and bradycardia
**Pickles CJ, 1989** [45]	Patients with mild to moderate, non-proteinuric pregnancy-induced hypertension treated with labetalol or placebo	*n* = 70 cases vs. *n* = 74 controls	England	RCT	The fetal outcome of labetalol versus placebo in pregnancy-induced hypertension	Pregnancy induced hypertension: a blood pressure of 140–160 mmHg systolic and 90–105 mmHg diastolic after 15 min rest on two occasions separated by 24 h	Hypoglycemia and bradycardia
**Ramanathan J, 1988** [28]	Woman with pre-eclampsia who were scheduled to undergo caesarean section under general anesthesia receiving labetalol pretreatmemt or no antihypertensive therapy before induction of anesthesia	*n* = 15 cases vs. *n* = 10 controls	United States	RCT	To study the effectiveness of labetalol in attenuating the hypertensive and tachycardiac responses associated with laryngoscopy and endotracheal intubation in pre-eclamptic women undergoing general anesthesia for caesarean section	Pre-eclampsia (diastolic blood pressure 96 to 120 mmHg and proteinuria) in combination with caesarean section	Hypoglycemia and bradycardia
**Ashe RG, 1987** [46]	Primigravida’s with severe hypertension in pregnancy at 32 weeks’ gestation or more receiving labetalol or dihydrallazine	*n* = 10 cases vs. *n* = 10 controls	South Africa	RCT	To compare the efficacy of dihydralazine with labetalol when administered as intravenous infusions to primigravida’s with severe hypertension in pregnancy at 32 weeks’ gestation or more	Severe hypertension in pregnancy (a diastolic blood pressure of 110 mmHg or more (Korotkoff phase IV sound), which had not settled after 2 h bed rest and sedation with phenobarbitone (sodium gardenal 200 mg intramuscularly))	Hypoglycemia
**Mabie WC, 1987** [47]	Pregnant women with hypertension during pregnancy or in the puerperium receiving labetalol or hydralazine	*n* = 40 cases vs. *n* = 20 controls	United States	RCT	To compare the safety and efficacy of intravenous labetalol and intravenous hydralazine hydrochloride for acutely lowering blood pressure in the pregnant or recently postpartum patient	Pre-eclampsia and chronic hypertension with or without superimposed pre-eclampsia	Hypoglycemia and bradycardia
**Boutroy MJ, 1986** [48]	Hypertensive mothers	*n* = 7	France	Case series	To evaluate the possible risk of exposure to beta blockers of newborn infants breast-fed by mothers being treated with acebutolol	Hypertension in pregnancy	Bradycardia
**Macpherson M, 1986** [49]	Infants born to women with hypertensive disease of pregnancy who had received labetalol for at least 7 days before delivery although some had begun treatment at 16 weeks gestation	*n* = 11 cases vs. *n* = 11 controls	England	Case–control	To examine a number of aspects of sympathetic function in infants born to labetalol-treated mothers compared with untreated controls to see if there were any clinically important effects of combined alfa and beta blockade	Hypertensive disease of pregnancy	Hypoglycemia and bradycardia
**Högstedt S, 1985** [50]	Women with mild and moderate hypertension in pregnancy treated with metoprolol and hydralazine vs. control	*n* = 82 cases vs. *n* = 79 controls	Sweden	RCT	To assess whether treatment with metoprolol, a beta-1 selective adrenoceptor blocking agent, in combination with hydralazine is of benefit for the mother and/or the fetus as compared with non-pharmacological treatment, in mild to moderate hypertension of pregnancy	A diastolic blood pressure of at least 90 mmHg on two or more occasions during pregnancy	Hypoglycemia and bradycardia
**Reynolds B, 1984** [51]	Women who developed hypertension in the last trimester of pregnancy received Atenolol or placebo	*n* = 60 cases vs. *n* = 60 controls	Scotland	RCT	To describe the findings of pediatric follow up to 1 year of age after the use of atenolol in pregnancy-associated hypertension	Pregnancy-associated hypertension	Bradycardia
**Williams ER, 1983** [52]	Women with mild to moderate hypertension	*n* = 9 Acebutolol and *n* = 11 Methyldopa	England	Case–control	To compare acebutolol with methyldopa in hypertensive pregnancy	A blood pressure of 130/90 mmHg or above, a systolic pressure of 135 mmHg or above or a diastolic pressure of 85 mmHg or above	Hypoglycemia and bradycardia
**Rubin PC, 1983** [53]	Women with mild to moderate pregnancy-associated hypertension who were also initially managed conventionally by bed rest received atenolol or placebo	*n* = 46 cases vs. *n* = 39 controls	Scotland	RCT	To examine the efficacy and safety of atenolol in the treatment of pregnancy-associated hypertension	Pregnancy-associated hypertension: a blood-pressure between 140 and 170 mmHg systolic or between 90 and 110 mmHg diastolic (after 10 mins’ rest supine or after 5 mins’ standing) on two occasions separated by 24 h	Hypoglycemia and bradycardia
**Liedholm H, 1983** [54]	Hypertensive pregnancies	*n* = 88 cases vs. *n* = 22 controls	Sweden	Cohort study	To determine the effects of atenolol and metoprolol on maternal blood pressure and on the fetus and new-born	Chronic or pregnancy-related hypertension	Hypoglycemia
**Liedholm H, 1983** [27]	Pregnant women under treatment with atenolol for hypertension (during pregnancy or in the peripartum)	*n* = 7	Sweden	Case series	To investigate atenolol’s ability to cross the human placental barrier and to study the excretion of atenolol in breast milk.	Hypertension in pregnancy	Bradycardia
**Livingstone I, 1983** [55]	Pregnancy-associated hypertension treated with propranolol or methyldopa	*n* = 14 cases and *n* = 14 controls	Australia	RCT	To compare propranolol with methyldopa in hypertensive pregnancy	A blood pressure of 140/90 or above, on two consecutive readings at least twenty-four hours apart.	Hypoglycemia and bradycardia
**Dubois D, 1983** [26]	High-risk pregnancies with hypertension using beta blockers	*n* = 125	France	Case series	To investigate the outcome of beta blocker use in high-risk pregnancies	Hypertension in pregnancy	Hypoglycemia
**Boutroy MJ, 1982** [56]	Children born from hypertensive pregnant women treated with acebutolol	*n* = 31	France	Case series	To determine the pharmacokinetics of acebutolol in the mother, as well as its placental transfer, and the pharmacokinetics in the fetus	Chronic or pregnancy-associated hypertension, after failure of strict bed rest and methyldopa with or without hydralazine	Bradycardia
**Rubin PC, 1982** [57]	Infants of women using atenolol for management of essential hypertension in pregnancy	*n* = 9	Scotland	Case series	To report the experience of using atenolol for several weeks during pregnancy in the management of essential hypertension	Systolic blood pressure exceeding 140 mmHg or diastolic pressure exceeding 90 mmHg on two separate occasions at least one day apart	Hypoglycemia and bradycardia
**Sandström B, 1982** [58]	Pregnant women with hypertension treated with metoprolol combined with thiazide or hydralazine compared with women treated with hydralazine and a thiazide	*n* = 184 cases (*n* = 101 with thiazide and *n* = 83 with hydralazine and *n* = 97 controls	Sweden	Case–control	To report further experiences of using metoprolol in hypertension of pregnancy. (In addition to a previous study)	Pregnancy-induced hypertension, pre-existing hypertension, eclampsia and hypertension with moderate/marked proteinuria	Hypoglycemia and bradycardia
**Garden A, 1982** [59]	Women with severe hypertension in pregnancy treated with labetalol or dihydralazine	*n* = 3 cases vs. *n* = 3 controls	South Africa	RCT	To compare the effect of labetalol and dihydralazine in increasing doses in woman with severe hypertension in pregnancy	Severe hypertension and imminent eclampsia or eclampsia	Bradycardia
**Dumez Y, 1981** [60]	Infants born to mothers who received acebutolol or methyldopa during pregnancy	*n* = 10 cases vs. *n* = 10 controls	France	Case–control	To evaluate any deleterious effect of the beta-adrenergic-blocking agent in newborn infants.	If the diastolic blood pressure exceeded 90 mm Hg on two occasions at least 24 h apart during pregnancy	Hypoglycemia and bradycardia
**Bott-Kanner G, 1980** [61]	Infants of mothers treated with propranolol and hydralazine because of longstanding hypertension during pregnancy	*n* = 14	Israel	Case series	To assess the efficiency of a combination of hydralazine and propranolol in the management of pregnant patients with essential hypertension	Essential hypertension	Hypoglycemia
**O’Hare MF, 1980** [62]	Hypertensive pregnant women receiving sotalol	*n* = 12	Northern Ireland	Case series	To study the effects and distribution of sotalol by administering it as sole therapy to a group of chronically hypertensive pregnant women.	Chronic or pregnancy-induced hypertension	Hypoglycemia and bradycardia
**Gallery ED, 1979** [63]	Pregnant women with moderately severe hypertension treated with oxprenolol or methyldopa	*n* = 26 cases vs. *n* = 27 controls	Australia	RCT	To examine the effects of antihypertensive treatment more closely and to evaluate alternative forms of treatment	Moderately severe hypertension in pregnancy	Hypoglycemia
**Pruyn SC, 1979** [64]	Infants from mothers who used propranolol chronically during pregnancy	*n* = 12	United States	Case series	To examine maternal, fetal and neonatal complications of propranolol therapy in pregnancy	Thyrotoxicosis, hypertension, Barlow syndrome with arrhythmia, Lown-Ganong-Levine syndrome and supraventricular/paroxysmal atrial tachycardia	Hypoglycemia and bradycardia
**Eliahou HE, 1978** [65]	Infants from mothers treated with propranolol during pregnancy	*n* = 22	Israel	Case series	To report the experience of 25 women who received propranolol orally for the treatment of hypertension during 26 pregnancies with 22 liveborn infants	Essential hypertension, recurrent hypertension of pregnancy, pre-eclampsia and unilateral chronic pyelonephritis	Hypoglycemia

Abbreviations: HDP: Hypertensive Disorders in Pregnancy, HELLP: Hemolysis, Elevated Liver enzymes and Low Platelets, RCT: Randomized Controlled Trial.

### 3.2. Risk of Bias Assessment

#### 3.2.1. RCTs

Amongst the 18 RCTs included, 15 were classified as having a high risk of overall bias [28,36,38,41,42,44,45,46,47,50,51,53,55,59,63], two had some concerns [33,40] and one low risk [29]. Most of the RCTs had some concerns for the domains of the randomization process since the process of allocation concealment was not mentioned, which describes deviations from the intended interventions and bias due to selection of the reported results with an absence of an a priori registered trial protocol. The risk of bias in the RCTs is shown in Table 4.

#### 3.2.2. Non-RCTs

Amongst the 13 non-RCTs, four had critical [32,39,54,60], six had serious [31,34,43,49,52,58] and three had a moderate risk of overall bias [30,35,37]. Most of the studies had issues of confounding, selection bias, bias in classification of interventions and selective reporting. Table 5 shows the risk of bias in non-randomized controlled trials. Table 6 shows the quality of the nine case series [26,27,48,56,57,61,62,64,65]. Table 7 shows the quality of the 15 case reports [66,67,68,69,70,71,72,73,74,75,76,77,78,79,80].

### 3.3. Bradycardia

Forty-one of the fifty-five included articles examined whether bradycardia was found in neonates exposed to beta-blockers either in utero or during lactation [27,28,30,32,33,34,35,37,38,40,43,44,45,47,48,49,50,51,52,53,55,56,57,58,59,60,62,64,67,68,69,70,71,72,73,74,75,76,77,78,79]. Table 8 presents 28 of the articles [27,28,30,32,33,34,35,37,38,40,43,44,45,47,48,49,50,51,52,53,55,56,57,58,59,60,62,64] that reported on the heart rate of neonates exposed to beta-blockers in utero. The other thirteen articles are case reports [67,68,69,70,71,72,73,74,75,76,77,78,79], which are described in Section 3.6. One case in the study of Boutroy et al. [48] had hypotension. However, the extent of the hypotension and the need for treatment was not described [48].

#### 3.3.1. Beta-Blocker vs. Control Group without a Beta-Blocker

Seven of the included articles [28,32,34,35,37,43,49] studied the heart rate of neonates exposed to a beta-blocker compared to a control group without beta-blocker exposure. These controls had mothers with hypertensive diseases and were exposed to antihypertensive drugs other than beta-blockers. However, in these studies not a specific other agent, such as methyldopa, was compared to the beta-blocker group. Two of these seven articles [32,35] showed that bradycardia occurred significantly more in neonates exposed to a beta-blocker in-utero. Mazkereth et al. [32] showed that 3.9% of the neonates of mothers treated with propranolol, labetalol or metoprolol experienced bradycardia, while bradycardia did not occur in the control group (*p* = 0.03). The six neonates with documented bradycardia were asymptomatic and were discharged following 24 h of non-bradycardic heart monitoring [32]. None of the neonates needed to be admitted to the neonatal intensive care unit [32]. Bateman et al. [35] observed a 30% increase in the risk of neonatal bradycardia in infants born to mothers using beta-blockers. The other five articles [28,34,37,43,49] showed no significant difference in the occurrence of bradycardia. In one of these studies the mother received labetalol solely as a short-term pretreatment for anesthesia prior to cesarean section [28].

#### 3.3.2. Beta-Blocker vs. Placebo

One of four studies comparing a beta-blocker exposed group with a placebo group observed that bradycardia occurred more often in neonates exposed to atenolol when compared with the placebo group (39.1 vs. 10.3 %, *p* < 0.01) [53]. In none of the cases was treatment for bradycardia needed [53]. The three other studies did not find a significant difference between the beta-blockers exposed group and the placebo group [45,51,81].

#### 3.3.3. Beta-Blocker vs. Methyldopa

Six articles [30,33,38,52,55,60] studied the difference in the heart rate of neonates exposed to either a beta-blocker or methyldopa in utero. One of these studies [60] found a significantly lower heart rate in neonates exposed to acebutolol when compared to those exposed to methyldopa (see Table 8) and therefore concluded that acebutolol has a long-lasting neonatal hemodynamic effect. However, another study comparing acebutolol to methyldopa found no signs of bradycardia in both groups [52].

**Table 8 ijerph-19-09616-t008:** Studies with bradycardia as outcome measure.

Article	Study Groups	Bradycardia	Heart Rate (Beats per Minute)	Definition of Bradycardia	Time of Control of Bradycardia
**Kayser A, 2020** [30]	ß-blocker vs. methyldopa ß-blocker vs. non-hypertensive mother	5/252 (2.0%) vs. 5/199 (2.5%) (NS) 5/252 (2.0%) vs. 5/588 (0.8%) (NS)		Diagnosis of bradycardia was retrieved from medical reports	N
**Mazkereth R, 2019** [32]	ß-blocker vs. control	6/153 (3.9%) vs. 0/153 (0%) *p* = 0.030		Heart rate < 100 bpm	In the first 48 h after birth
**Easterling T, 2019** [33]	Labetalol vs. nifedipine Labetalol vs. methyldopa	0/280 (0%) vs. 0/297 (0%) 0/280 (0%) vs. 0/298 (0%)		Heart rate < 110 bpm	N
**Thewissen L, 2017** [34]	Labetalol vs. control	0/22 vs. 0/22 vs. 0/22		N	In the first 24 h after birth
**Bateman BT, 2016** [35]	ß-blocker vs. control Labetalol vs. control Metoprolol vs. control Atenolol vs. control Propensity-score ^1^ corrected available	165/10,585 (1.6%) vs. 11,659/2,281,531 (0.5%) 124/6748 (1.8%) vs. 11,659/2,281,531 (0.5%) 12/1485 (0.8%) vs. 11,659/2,281,531 (0.5%) 12/1121 (1.1%) vs. 11,659/2,281,531 (0.5%)		Heart rate ≤ 100 bpm	N
**Heida KY,** **2012** [37]	Labetalol vs. control Labetalol i.v. vs. labetalol oral	4/55 (7.3%) vs. 1/54 (1.9%) *p* = 0.18 5.4 vs 11.1% *p* = 0.39		Heart rate < 100 bpm	In the first minutes after birth and during the first 48 h
**Verma R, 2012** [38]	Labetalol vs. methyldopa	1/45 (2.22%) vs. 0/45 (0%) (NS)		N	N
**Vigil-De Gracia P, 2006** [40]	Labetalol vs. hydralazine	11/103 (10.6%) vs. 2/102 (1.9%) (*p* = 0.008)		Heart rate < 110 bpm	N
**Munshi UK, 1992** [43]	Labetalol vs. control	6/48 (12.5%) vs 4/81 (5%) (NS)		Heart rate < 100 bpm	At 5-min as part of Apgar scoring
**Bott-Kanner G, 1992** [44]	Pindolol vs. placebo	^2^		Heart rate < 100 bpm	During the first 24 h after birth
**Pickles CJ, 1989** [45]	Labetalol vs. placebo	4/70 (5.7%) vs. 4/74 (5.4%)		Heart rate < 120 bpm	At five minutes
**Ramanathan J, 1988** [28]	Labetalol vs. control	0/15 (0%) vs. 0/10 (0%)	138.2 ± 2.5 vs. 144 ± 3.2 (NS)	N	During 10–20 min after birth and thereafter for 12 to 24 h
**Mabie WC, 1987** [47]	Labetalol vs. hydralazine	0/13 (0%) vs. 0/6 (0%)		Heart rate < 110 bpm	N
**Macpherson M, 1986** [49]	Labetalol vs. control		No difference between the two groups	N	At 2, 4, 8,16, 24, 48 and 72 h after birth
**Boutroy MJ, 1986** [48]	^3^				
**Högstedt S, 1985** [50]	Metoprolol and hydralazine vs. control (intended-to-treat) ^4^ Metoprolol and hydralazine vs. control (cause–effect)	1/82 (1.2%) vs. 4/79 (5.1%) 1/69 (1.4%) vs. 3/66 (4.5%)		N	N
**Reynolds B, 1984** [51]	^5^				
**Williams ER, 1983** [52]	Acebutolol vs. methyldopa	0/9 vs. 0/11		N	N
**Rubin PC, 1983** [53]	Atenolol vs. placebo	18/46 (39.1%) vs. 4/39 (10.3%) (*p* < 0.01)		Heart rate < 120 bpm	Continuously recorded in the first 24 h after birth
**Liedholm H, 1983** [27]	Atenolol (no control group)	^6^		N	N
**Livingstone I, 1983** [55]	Propranolol vs. methyldopa	0/14 (0%) vs. 0/14 (0%)		N	During 48 h after birth
**Boutroy MJ, 1982** [56]	Acebutolol (no control group)	12/31 (38.7%)		Basal heart rate < 120 beats per minute and lasting longer than 1 h	During 72 h after birth
**Sandström B, 1982** [58]	Bendroflumethiazide + metoprolol vs. metoprolol + hydralazine vs. Bendroflumethiazide + hydralazine	7/101 vs. 1/83 vs. 16/97 (8/184 vs. 16/97)		Heart rate < 100 bpm	At birth
**Garden A, 1982** [59]	Labetalol vs. hydralazine	3/3 (100%) vs. 0/3 (0%)		Heart rate < 100 bpm	Immediately after birth
**Rubin PC, 1982** [57]	Atenolol (no control group)	0/9 (0%)		Heart rate < 120 bpm	During 24 h after birth
**Dumez Y, 1981** [60]	Acebutolol vs. methyldopa		Day 1: 118 ± 19 vs. 132 ± 9 (*p* < 0.05) Day 2: 123 ± 18 vs. 139 ± 10 (*p* < 0.05) Day 3: 126 ± 21 vs. 148 ± 12 (*p* < 0.02)	N	Daily during the 3 first days after birth, when the babies were sleeping
**O’Hare MF, 1980** [62]	Sotalol (no control group)	6/12 (50%)		Heart rate < 120 bpm	Four-hourly for at least 24 h
**Pruyn SC, 1979** [64]	Propranolol (no control group)	1/12 (8.3%)		N	N

Abbreviations: N: is not described in article; NS: not significant; bpm: beats per minute. ^1^ PS-matched: Propensity scores were estimated using a logistic regression model in which exposure was the dependent variable and was estimated on the basis of 5 groups of potential confounders of the planned analysis: demographic characteristics, medical conditions, obstetrical conditions, maternal medications, and measures of healthcare use [35]. ^2^ As regards other outcome variables, namely, the Apgar score, respiratory and heart rate at delivery, hypoglycemia and jaundice during the first 24 h—the differences between the two treatment groups were inconsistent and non-significant [44]. ^3^ Hypotension, bradycardia and transient tachypnea were observed in one infant. The article does not describe if there were any other cases of bradycardia. ^4^ For the analyses, the material was divided into two categories. The first group gives data for all the 161 patients whom it was the intention to treat. In the calculation of cause-and-effect, 26 patients were withdrawn from the original group of 161: in 5 patients of C-group, DBP exceeded 110 mmHg and they were then treated with antihypertensive drugs; one patient in T-group admitted that she had not taken the prescribed drugs; 6 patients gave birth to malformed or stillborn children and their data were not used for the calculation of Apgar scores, birth weights or other vitality signs. Eight patients in the T-group and 6 in the C-group gave birth within 2 weeks after admission to the study, and these 14 women were excluded from the cause-and-effect analyses because of the short treatment period [50]. ^5^ One infant in the placebo group had a bradycardia in the first 12 h of life. The article does not describe if there were any other cases of bradycardia. ^6^ Bradycardia was only investigated in one of the seven infants [27]. At no time did this infant have bradycardia or any other clinical sign of beta-blockade [27].

#### 3.3.4. Beta-Blocker vs. Hydralazine

Three articles [40,47,59] studied the effects of beta-blockers compared to hydralazine on the heart rate of neonates after being exposed in utero. Vigil et al. [40] observed a significantly higher rate of bradycardia in neonates exposed to labetalol compared with neonates exposed to hydralazine. In the study of Garden et al. [59] all three infants in the labetalol group were born with a bradycardia compared to none of the three neonates who were exposed to hydralazine. All three infants in the labetalol group were growth-retarded [59]. On the other hand, Mabie et al. [47] found no differences between both groups (40 cases labetalol vs. 20 hydralazine controls) regarding neonatal bradycardia.

#### 3.3.5. No Control Group

Three articles [27,56,62] studied the effect of in utero exposure to beta-blockers on neonatal heart rate without comparison to a control group. Boutroy et al. [56] observed bradycardia, defined as a basal heart rate of less than 120 beats/minute and lasting longer than one hour, in twelve of the 31 neonates exposed to atenolol. O’Hare et al. [62] detected bradycardia, also defined as less than 120 beats/minute, in six of the twelve neonates who were exposed to sotalol in utero. The heart rate in these neonates was in a range from 90 to 120 beats/minute and stayed present for up to 25 h after birth in five neonates [62]. The neonates did not suffer from any negative symptoms of the bradycardia [62]. Another study [64] noted bradycardia in one of the twelve neonates exposed to propranolol in utero. This neonate was born only 29 min after the last dose of propranolol [64].

#### 3.3.6. Meta-Analyses for the Outcome Bradycardia

Funnel plot evaluation and Egger’s tests of the included RCTs in the meta-analyses for the outcome bradycardia indicated no publication bias (Figure 2a). The regression test however indicated publication bias for the proportion-based meta-analysis for bradycardia (Figure 2b). Beta-blockers possibly resulted in a higher risk of bradycardia when compared to other drugs or placebo or no therapy as assessed by the meta-analysis of RCTs (Relative risk (RR), 95% Confidence Interval (CI): 2.36, 1.33–4.18), with the CoE being low (Figure 3, Table 9). A sub-group analysis of RCTs revealed that the risk of bradycardia was possibly higher with beta-blockers when compared to placebo or no therapy (RR, 95% CI: 2.22, 1.07–4.64) and hydralazine (RR, 95% CI: 6.66, 1.61–27.57), but possibly similar when compared to methyldopa (RR, 95% CI: 11.0, 0.02–7032.32) and calcium channel blockers (RR not calculable due to zero events) (Figure 3). Clinical benefit or harm could not be excluded for the comparison beta-blockers versus placebo, no therapy or other drugs, as assessed from results of meta-analyses of cohort studies and case–control studies, because the estimates were statistically non-significant and the CoE was very low to low (Figure 4, Table 9). However, sub-group analyses of cohort studies (RR, 95% CI: 3.03, 2.60–3.53) and case–control studies (RR, 95% CI: 4.01, 1.50–10.75) showed that beta-blockers were possibly associated with higher risk of bradycardia when compared to placebo or no therapy, and a possibly similar risk when compared to methyldopa (Figure 4). The proportion-based meta-analysis showed that the overall incidence of bradycardia with beta-blockers was 6% (95% CI: 2%–13%) with statistically significant difference in the incidence between the various types of beta-blockers (*p* = 0.002) (Figure 5). While propranolol (2% (95% CI: 0%–16%)), metoprolol combined with hydralazine (3% (95% CI: 1–6%)) and labetalol (4% (95% CI: 0%–11%)) had the lowest risk, sotalol had the highest risk of possibly being associated with bradycardia (50% (95% CI: 22%–78%)).

### 3.4. Hypoglycemia

Forty-seven out of the 55 included articles investigated the occurrence of hypoglycemia in neonates exposed to beta-blockers either in utero or during lactation [26,28,29,30,31,32,33,35,36,37,38,39,40,41,42,43,44,45,46,47,49,50,52,53,54,55,57,58,60,61,62,63,64,65,66,67,68,70,72,73,74,75,76,77,78,79,80]. Table 10 shows 34 of the articles reporting hypoglycemia and the definitions used by the authors [26,28,29,30,31,32,33,35,36,37,38,39,40,41,42,43,44,45,46,47,49,50,52,53,54,55,57,58,60,61,62,63,64,65]. The other 13 articles investigating hypoglycemia are case reports and are described in Section 3.6.

#### 3.4.1. Beta-Blocker vs. Control Group without a Beta-Blocker

Eight articles [28,32,35,37,39,41,43,49] studying the blood glucose levels in neonates compared a group exposed to beta-blockers with a control group of neonates non-exposed to beta-blockers in utero. Five out of these eight articles [32,35,39,41,43] showed a significantly higher risk for hypoglycemia in neonates exposed to beta-blockers in utero (Table 10). For example, in the article of Mazkereth et al. [32] hypoglycemia occurred more often in the beta-blocker exposed neonates than in the non-exposed neonates (30.7% vs. control 18.3%, *p* = 0.016). Darcie et al. [41] showed that 65% of the neonates who were exposed to beta-blockers in utero developed hypoglycemia when compared to only 28,5% of the non-exposed neonates in the control group (*p* < 0.05). Kumar et al. [31] found out that the odds of neonatal hypoglycemia are 1.75 times higher when the baby was exposed to beta-blockers in utero. Kumar et al. [31] reported that the risk for developing neonatal hypoglycemia was 3.15 times higher when the maternal beta-blocker usage was combined with maternal diabetes during pregnancy. The study of Ramanathan et al. [28] did not find an increased risk for hypoglycemia. However, in this study labetalol was prescribed for short-term use solely for the induction of anesthesia [28].

#### 3.4.2. Beta-Blocker vs. Placebo

Four articles [29,44,45,53] compared neonates exposed in utero to beta-blockers with neonates exposed to a placebo. None of these articles found a significant difference regarding hypoglycemia between the two study groups. However, in the study of Bigelow et al. [29] mothers received propranolol solely short term for the induction of labor.

#### 3.4.3. Beta-Blocker vs. Methyldopa

Seven articles [30,33,38,52,55,60,63] studied neonates born after in utero exposure to either a beta-blocker or methyldopa. Gallery et al. [63] showed that the blood sugar levels of neonates exposed to methyldopa in utero were significantly lower than those of neonates exposed to oxprenolol. None of the neonates in the oxprenolol group of this study were clinically hypoglycemic, meanwhile two neonates in the methyldopa group were clinically hypoglycemic [63]. The studies of Verma and Easterling et al. [33,38] showed no significant differences in neonatal hypoglycemia after labetalol or methyldopa exposure in utero.

#### 3.4.4. Beta-Blocker vs. Hydralazine

Four articles [36,40,46,47] studied the difference in effect of in utero exposure to a beta-blocker and in utero exposure to hydralazine on neonates. None of these articles found a significant difference in occurrence of hypoglycemia in these neonates [36,40,46,47]. In the study of Ashe et al. [46] one neonate was diagnosed with hypoglycemia; however, this neonate was severely growth-retarded, which is a known risk factor for hypoglycemia.

#### 3.4.5. No Control Group

Seven articles [26,54,57,61,62,64,65] examined the effect of beta-blockers on the neonate without comparing with a control group. Dubois et al. [26] found one neonate with a low cord blood glucose level who was exposed to acebutolol in utero. Another study done by O’Hare et al. [62] showed that one of the small for gestational age (SGA) infants suffered an episode of hypoglycemia one hour after birth after being exposed to sotalol in utero. This neonate responded well to a single dose of dextrose [62]. They found no additional symptoms suggesting an adverse effect of sotalol in the twelve infants included in the study and therefore did not adjust or discontinue the use of sotalol by the mothers [62]. In the study of Pruyn et al. [64], three of the twelve neonates who were in utero exposed to propranolol were diagnosed with hypoglycemia. One of those neonates was small for their gestational age, one was delivered by Cesarean section for fetal distress and one was the result of a precipitous delivery 29 min after the last dose of propranolol [64]. Eliahou et al. [65] measured the blood sugar in thirteen of the neonates who were in utero exposed to propranolol and these were all normal.

#### 3.4.6. Meta-Analyses for the Outcome Hypoglycemia

A moderate CoE from meta-analyses of cohort studies indicated that beta-blockers were probably associated with a significantly higher risk of hypoglycemia when compared to placebo, no therapy or other drugs (RR, 95% CI: 3.01, 2.79–3.25) (Figure 6, Table 9). A sub-group analysis also revealed that beta-blockers were possibly associated with a higher risk of hypoglycemia when compared to placebo or no therapy (RR, 95% CI: 3.05, 2.82–3.29) and calcium channel blockers (RR, 95% CI: 2.42, 1.47–4.00), but not when compared with methyldopa (RR, 95% CI: 1.84, 0.48–7.03) (Figure 6).

Quite like the results from the cohort studies, the meta-analysis of case–control studies indicated that beta-blockers were possibly associated with a higher risk of hypoglycemia when compared to placebo, no therapy or other drugs (RR, 95% CI: 1.72, 1.133–2.22), with the CoE being very low (Figure 6, Table 9). The sub-group analysis indicated a possibly higher risk of hypoglycemia with beta-blockers when compared to placebo or no therapy (RR, 95% CI: 1.68, 1.03–2.73), but not when compared with methyldopa (RR, 95% CI: 6.00, 0.87–41.21) (Figure 6). While the funnel plot of the included RCTs and the Egger’s test indicated no publication bias, we did detect publication bias for the proportion-based meta-analysis based on the regression test (Figure 7a,b). Clinical benefit or harm could not be excluded for the outcome of hypoglycemia from the results of the meta-analysis of RCTs, due to the result being statistically non-significant and the CoE being low (Figure 8, Table 9). The proportion-based meta-analysis suggested that the incidence of hypoglycemia with the use of any beta-blocker might be 12% (95% CI: 7–19%), with it being lowest with metoprolol (3% (95% CI: 2–4%)) and labetalol (8% (95% CI: 2–16%)) (Figure 9). Atenolol had the highest association with hypoglycemia (35% (95% CI: 3–77%)).

### 3.5. Lactation

Most of the included studies reported no specific information regarding the type of feeding of the neonate. Only the study by Kumar et al. [31] included the type of feeding into their multiple regression analysis. Interestingly, they showed that formula feeding was a risk factor for hypoglycemia (*p* < 0.001) and not breast milk [31]. Moreover, Boutroy et al. [48] described a case with symptoms of bradycardia and hypotension. This child was exposed to high concentrations of acebutolol through breast milk [48]. Another study in children exposed to sotalol showed no bradycardia in the child with the highest concentration in breast milk [62]. In the study of Liedholm et al. [27] no signs of beta blockade were observed after exposure to atenolol. O’Hare [62] and Liedholm [27], found a ratio of maternal plasma concentrations versus breast milk concentrations of 1:5.4 (sotalol) and 1:4.5 (atenolol). Although the concentrations in breast milk are higher than in maternal plasma, the corresponding expected serum concentrations in infants would be less than the daily dose for hypertensive patients in general [27,58]. However, according to O’Hare et al. [62] the expected plasma concentration in a breastfed infant for sotalol is expected to be within the therapeutic dose range.

### 3.6. Case Reports

Fifteen case reports [66,67,68,69,70,71,72,73,74,75,76,77,78,79,80] were included in this systematic review (Table 11). Eleven case reports [67,68,70,72,73,74,75,76,77,78,79] reported the occurrence of both hypoglycemia as well as bradycardia in a neonate. Two other articles [66,80] reported hypoglycemia in a neonate exposed to a beta-blocker in utero. Moreover, two other case reports [69,71] reported bradycardia after beta-blocker exposure in utero.

## 4. Discussion

There is an ongoing debate as to whether exposure to beta-blockers in utero and through lactation negatively affects the neonate, whereas beta-blockers are often used in pregnant women with cardiovascular diseases. As it is unknown to what extent beta-blockers harm the neonate, best clinical practice about heart rate and glucose monitoring is inconclusive. In this systematic review and meta-analysis, the occurrence of the postnatal neonatal side effects of hypoglycemia and bradycardia among neonates exposed to beta-blockers in utero or during lactation were evaluated systematically. Our overall aim was to assess the need of postnatal monitoring and observation of the neonate. Moreover, differences in neonatal risk between the different types of beta-blockers were studied. To the best of our knowledge, this is the first systematic review and meta-analysis evaluating the neonatal outcomes associated with fetal beta-blocker exposure.

Our meta-analysis showed that in utero exposure to beta-blockers possibly results in a higher risk for neonatal bradycardia. While our sub-group analysis of RCTs revealed that the risk of bradycardia was possibly higher with beta-blockers when compared to placebo or no therapy, this risk was possibly similar when compared to methyldopa, suggesting other causal factors for bradycardia rather than the beta-blocker exposure. Another explanation could be that methyldopa might be associated with neonatal bradycardia as well. Bradycardia has been reported as a side-effect of methyldopa with an unknown incidence [81]. This possibly explains the lack of difference between both groups of antihypertensive drugs. Future studies regarding the effect of the maternal use of methyldopa on the neonate are needed. However, clinical harm could not be excluded for the use of beta-blockers. Importantly, even if beta-blockers induce a higher risk for neonatal bradycardia, the clinical relevance of this in terms of hypotension and the required treatment and hospital stay need to be elaborated further in future studies. The treatment for resulting hypotension was not described in any of the reports and neither were other serious adverse events. The majority of case reports reported bradycardia after beta-blocker exposure. However, there is a possibility of publication bias being associated with evidence from case reports.

Regarding hypoglycemia, our meta-analysis indicated that in utero exposure to beta-blockers was probably associated with a significantly higher risk of hypoglycemia when compared to placebo, no therapy or other drugs than beta-blockers. However, no difference was found in comparison to methyldopa exposure. Yet again, the effects of methyldopa on the neonate need to be studied. Since beta-blocker exposure was probably associated with a higher risk for hypoglycemia, which is dangerous if untreated, we do suggest blood glucose monitoring during the first 24 h after birth for all beta-blockers.

Regarding the different types of beta-blockers, propranolol, metoprolol (combined with hydralazine) and labetalol had the lowest risk for bradycardia and sotalol had the highest risk of possibly being associated with bradycardia in neonates. Moreover, metoprolol and labetalol had the lowest risk for hypoglycemia and atenolol had the highest risk of possibly being associated with hypoglycemia in neonates. Fortunately, labetalol and metoprolol are more often prescribed than sotalol and atenolol during pregnancy and lactation [1]. Fetal and neonatal drug exposure and effects following maternal pharmacotherapy are influenced by multiple drug specific characteristics, i.a. lipophilicity, neonatal half-life, different dosages, the duration of maternal exposure as well as via lactation, the aspects of placental passage of beta-blockers, intra- and inter-patient variability in disposition, and their varying potency on neonatal bradycardia and hypoglycemia. Generally, fetal and neonatal exposure and effects are increased for drugs with a longer neonatal elimination half-life, more lipophilic drugs (leading to increased placental passage as well as disposition to breastmilk), increased dosages, and an increased duration of maternal treatment. This profile has not yet been reported for each beta-blocker used in clinical practice and requires further investigation. Moreover, future research may focus on neonatal effects after exposure to beta-blockers through lactation, as available literature is sparse.

The strength of this systematic review and meta-analysis is its clinically relevant scope along with the use of a robust search strategy, assessment of the overall CoE with the GRADE system, rating risks of biases using standard tools and conducting meta-analyses for the studied outcomes. However, this review has its limitations. Firstly, the definitions used for bradycardia and hypoglycemia were widely variable between the studies. For example, some studies defined bradycardia as a heart rate of less than 100 beats per minute while others used 120 beats per minute as the cut-off. Some other studies did not mention the definition at all. The same yielded true for the cut-off value for defining hypoglycemia. Unfortunately, since most studies only provided the occurrence of bradycardia or hypoglycemia instead of the raw data, we were unable perform any meta-regression. Secondly, only a few studies provided data on lactation. It could be that exposure through breastfeeding contributed to the positive findings in some studies as transmission of various beta-blockers to breastmilk has been reported [7,82,83,84]. Finally, we were unable to take into account the time of initiation of beta-blocker treatment, the total dose and duration of exposure, the exact indication and different gestational ages of the neonates due to lack of information.

## 5. Conclusions

Our meta-analysis determined a probable risk of hypoglycemia and possible risk of bradycardia in neonates upon neonatal beta-blocker exposure in utero or through lactation. As the CoE was moderate for the outcome of hypoglycemia, we suggest monitoring glucose levels in beta-blocker-exposed neonates until 24 h after birth irrespective of the type of used beta-blocker. Monitoring of the heart rate could be considered for 24 h, although the clinical implication of this needs to be evaluated in future studies. The necessity for heart rate monitoring may be adjusted according to the type of beta-blocker neonates are exposed to, which is most relevant for sotalol, and less for propranolol and labetalol.

## Figures and Tables

**Figure 1 ijerph-19-09616-f001:**
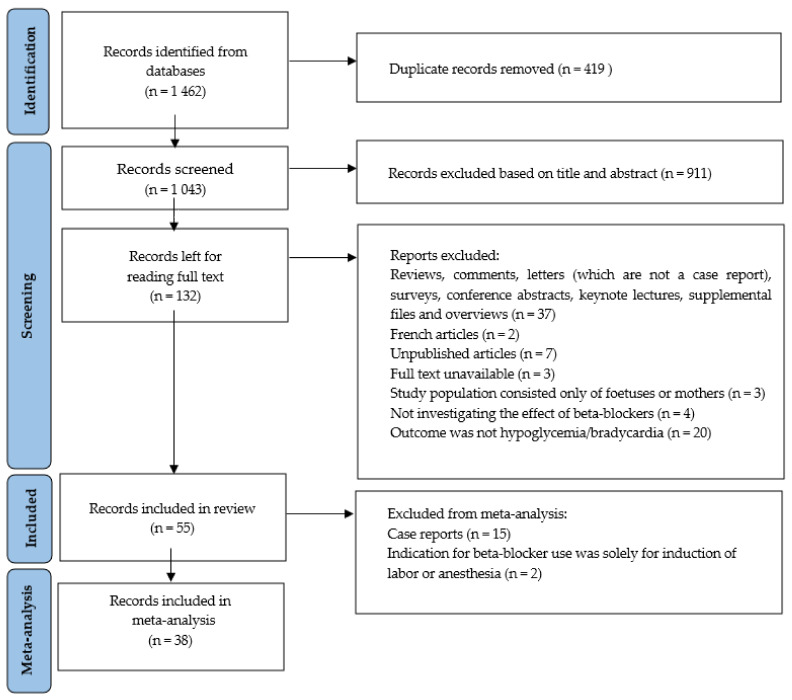
Overview literature search according the PRISMA guidelines.

**Figure 2 ijerph-19-09616-f002:**
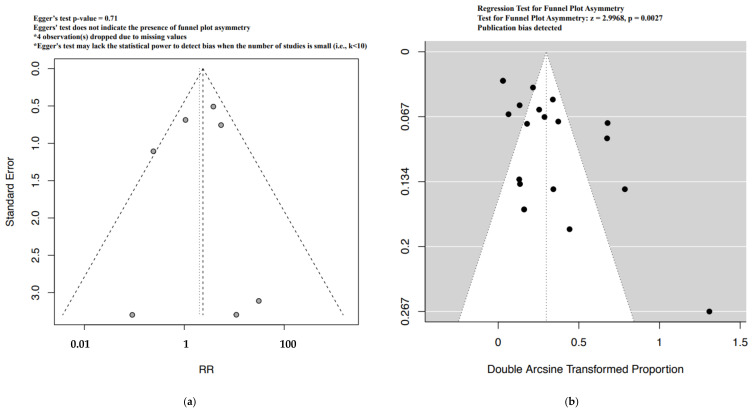
Funnel plots bradycardia—RCT (**a**) and proportion based meta-analysis (**b**).

**Figure 3 ijerph-19-09616-f003:**
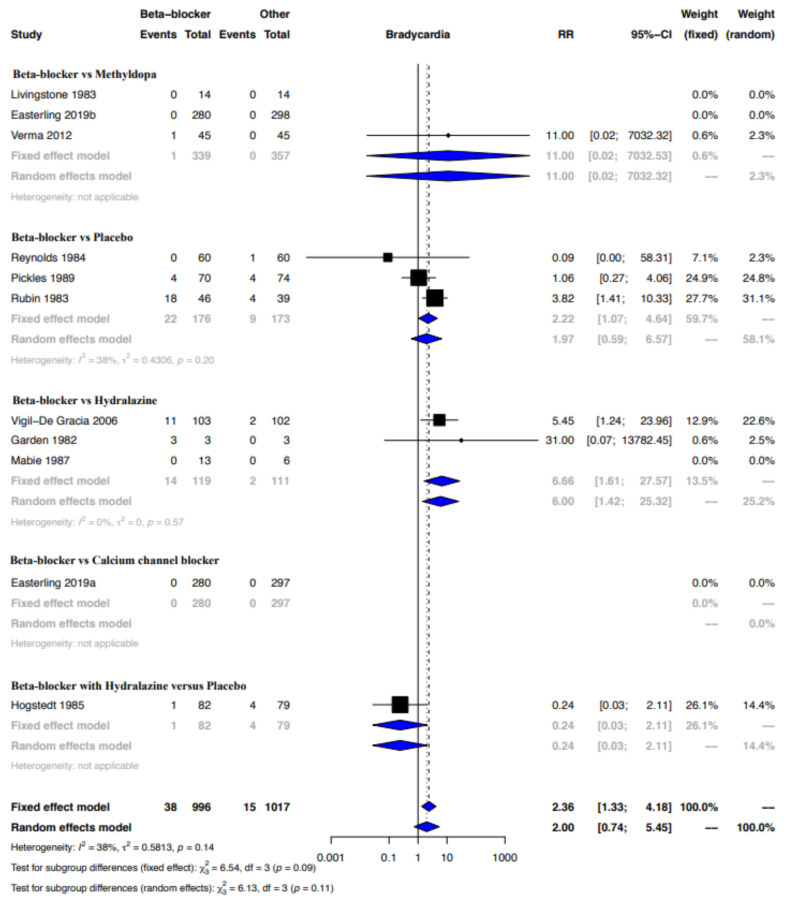
Meta-analysis bradycardia—RCT. Beta-blocker vs. methyldopa: [27,33,38]; beta-blocker vs. placebo: [45,51,53]; beta-blocker vs. hydralazine: [40,47,59]; beta-blocker vs. calcium channel blocker [33]; beta-blocker with hydralazine vs. placebo: [50].

**Figure 4 ijerph-19-09616-f004:**
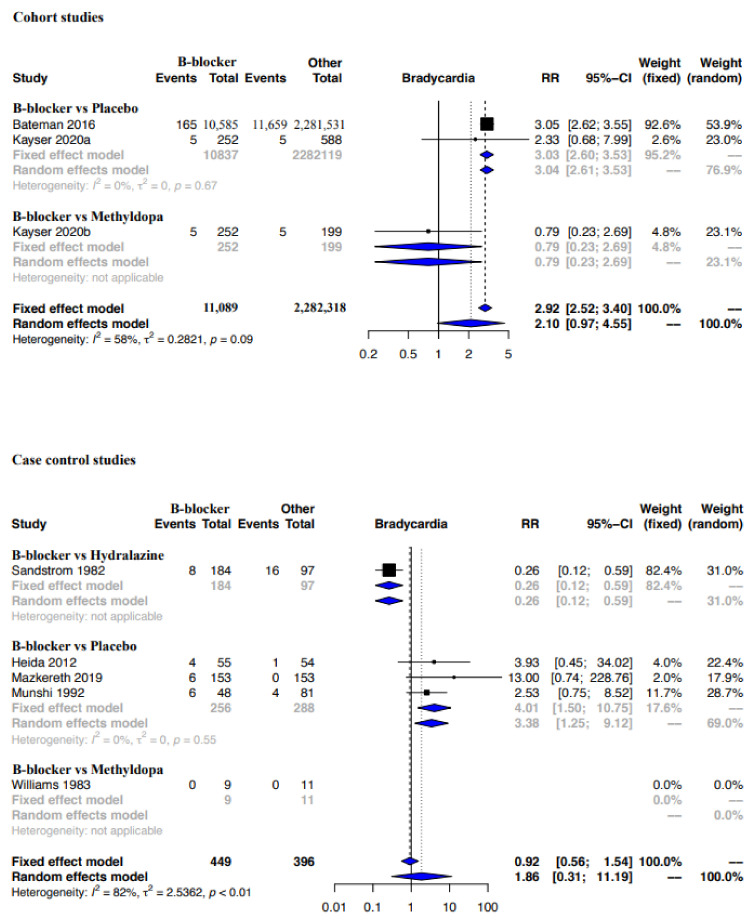
Meta-analysis bradycardia—Cohort and case–control studies. Cohort studies: beta-blocker vs. placebo: [30,35]; beta-blocker vs. methyldopa: [30]. Case control studies: beta-blocker vs. hydralazine: [58]; beta-blocker vs. placebo: [32,37,43]; beta-blocker vs. methyldopa: [52].

**Figure 5 ijerph-19-09616-f005:**
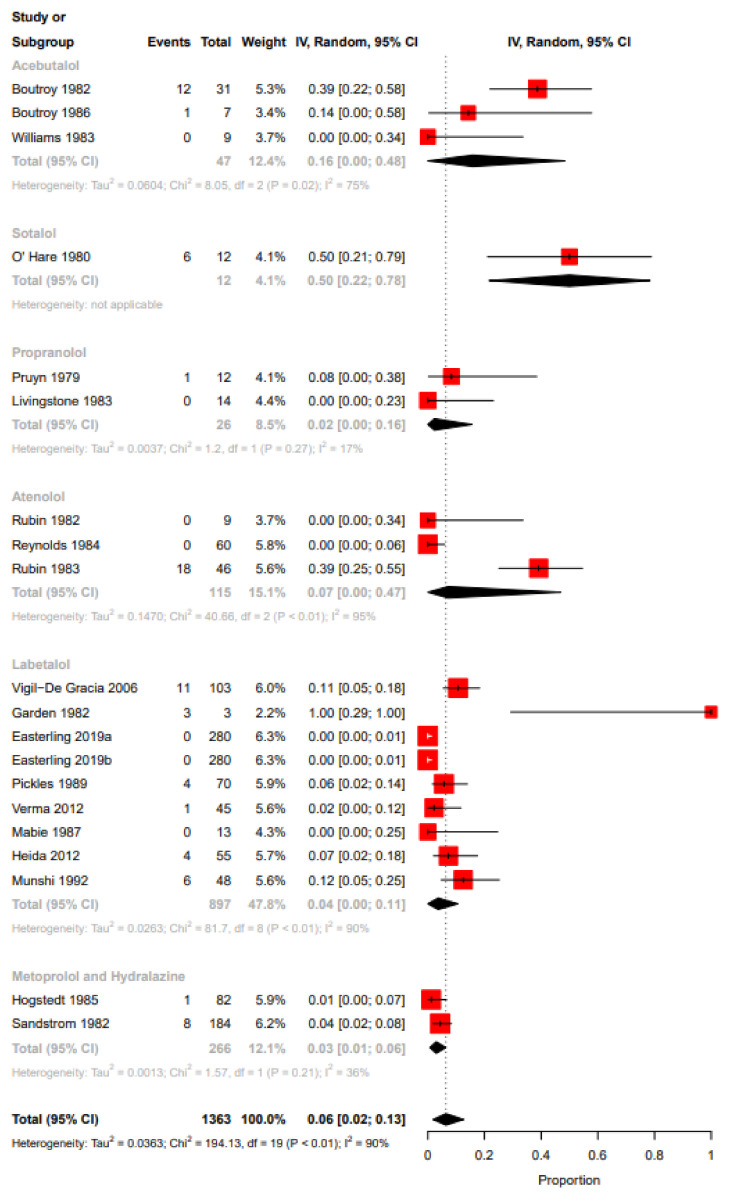
Proportion-based meta-analysis bradycardia. Acebutalol: [48,52,56]; sotalol: [62]; propranolol: [55,64]; atenolol: [51,53,57]; labetalol: [33,37,38,40,43,45,47,59]; metoprolol and hydralazine: [50,58].

**Figure 6 ijerph-19-09616-f006:**
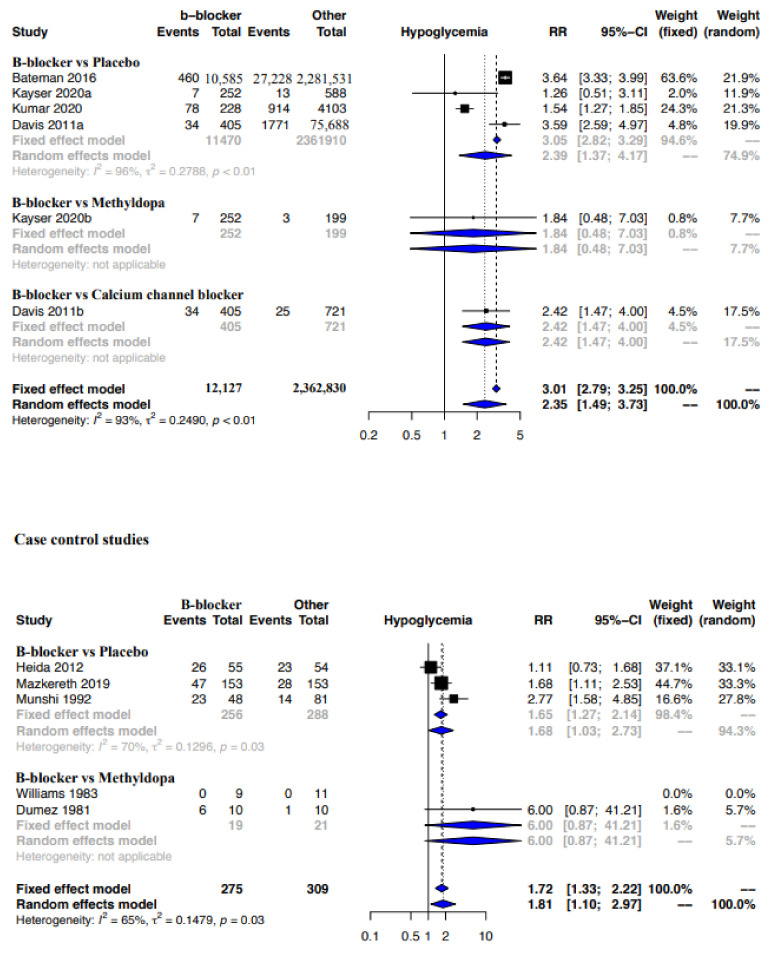
Meta-analysis hypoglycemia—Cohort and case–control studies. Cohort studies: beta-blockers vs. placebo: [30,31,35,39]; beta-blockers vs. methyldopa: [30]; beta-blockers vs. calcium channel blocker: [39]. Case control studies: beta-blocker vs. placebo: [32,37,43]; beta-blockers vs. methyldopa: [52,60].

**Figure 7 ijerph-19-09616-f007:**
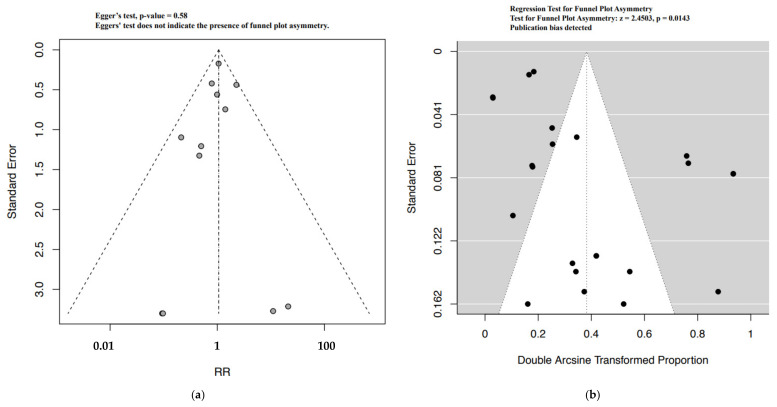
Funnel plots hypoglycemia—RCT (**a**) and proportion based meta-analysis (**b**).

**Figure 8 ijerph-19-09616-f008:**
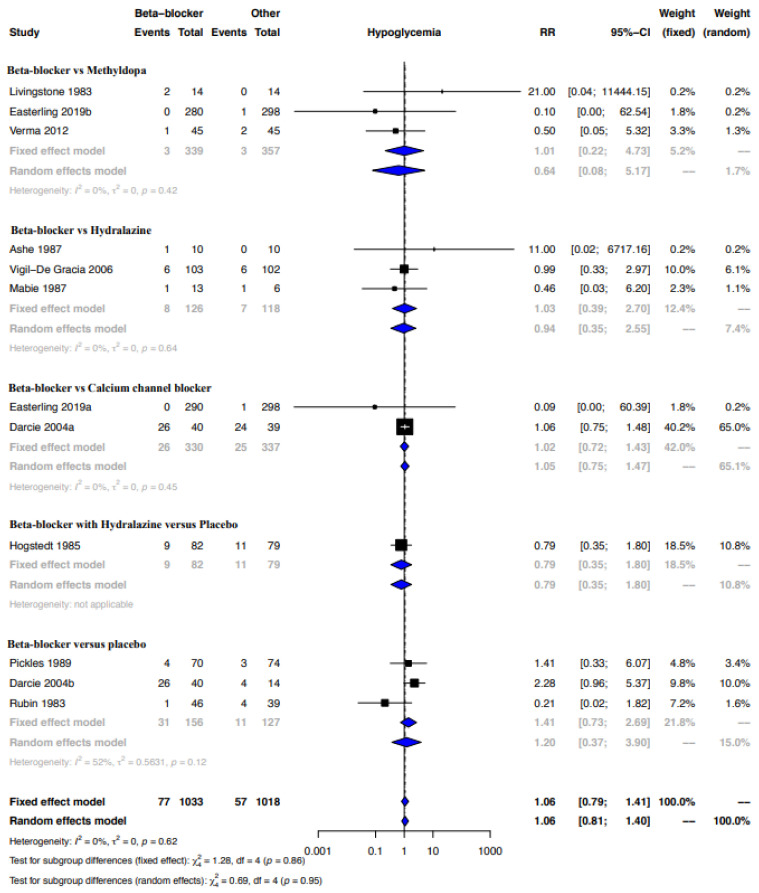
Meta-analysis hypoglycemia—RCT. Beta-blocker vs. methyldopa: [33,38,55]; beta-blockers vs. hydralazine: [40,46,47]; beta-blockers vs. calcium channel blockers: [33,41]; beta-blocker with hydralazine vs. placebo: [50]; beta-blockers vs. placebo [41,45,53].

**Figure 9 ijerph-19-09616-f009:**
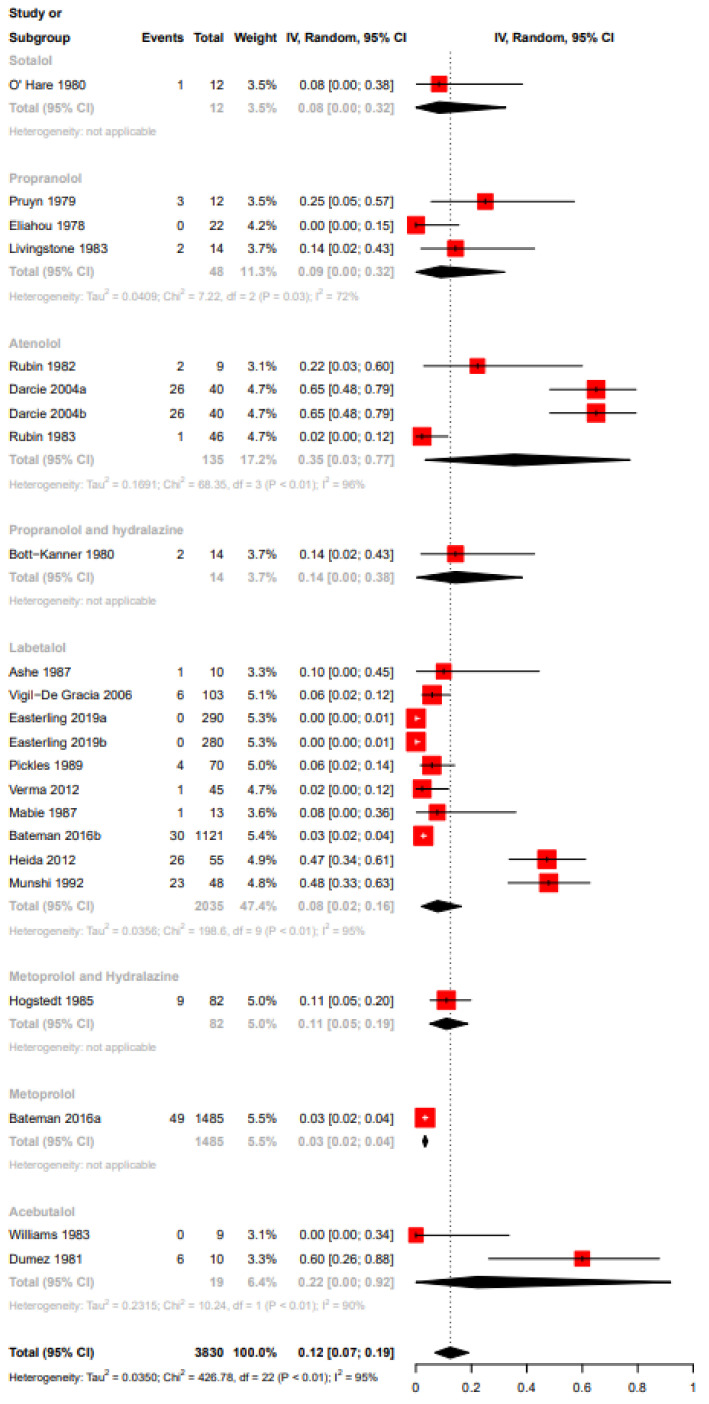
Proportion-based meta-analysis of hypoglycemia. Sotalol: [62]; propranolol: [55,64,65]; atenolol: [41,53,57]; propranolol and hydralazine: [61]; labetalol: [33,35,37,38,40,43,45,46,47]; metoprolol and hydralazine: [50]; metoprolol: [35]; acebutalol: [52,60].

**Table 1 ijerph-19-09616-t001:** Statements to communicate the findings of the systematic review.

Descriptor	Criteria
**Clinical** **benefit/harm**	Statistically significant result High certainty evidence Biological mechanism(s) well established Point estimates of underlying studies are consistently in one direction Optimal information size reached
**Probable clinical benefit/harm**	Statistically significant result Moderate or high certainty evidence Evidence of biological plausibility Point estimates of underlying studies are predominantly in one direction Close to optimal information size or summary confidence interval is sufficiently narrow to give confidence that the true effect would be clinically meaningful if it is only in the ballpark of the summary estimate
**Possible clinical benefit/harm**	Statistically significant result Low or very low certainty evidence Few studies, wide summary confidence interval or effect is driven by one or two heavily weighted studies
**Improbable** **benefit/harm**	Statistically non-significant result Moderate or high certainty evidence Point estimates of underlying studies are close to and on both sides of the line of null effect
**No clinical** **benefit/harm**	Statistically non-significant result High certainty evidence Point estimates of underlying studies are close to and on either side of the line of null effect Majority of underlying studies are adequately powered for outcome of interest Optimal information size reached
**Clinical benefit/harm cannot be excluded**	Statistically non-significant result Low or very low certainty evidence Few studies Wide confidence intervals

**Table 3 ijerph-19-09616-t003:** Type and dosage of beta blocker.

Article	Acebutolol	Atenolol	Bisoprolol	Carvedilol	Labetalol	Metoprolol	Nadolol	Oxprenolol	Pindolol	Propranolol	Sotalol
**Bigelow CA, 2021** [29]										2 mg	
**Kayser A, 2020** [30]			1.25–10 mg/day			12–400 mg/day					
**Kumar N, 2020 ^1,2^** [31]											
**Mazkereth R, 2019 ^2^** [32]					X	X				X	
**Easterling T, 2019** [33]					200 mg/day (max. 600 mg/day)						
**Thewissen L, 2017** [34]					X						
**Bateman BT, 2016 ^2^** [35]	X	X	X	X	X	X	X		X	X	X
**Singh R, 2016 ^3^** [36]					X						
**Heida KY,2012 ^4^** [37]					X						
**Verma R, 2012** [38]					100–300 mg three times per day						
**Davis RL, 2011 ^5^** [39]											
**Vigil-De Gracia P,** **2006 ^6^** [40]					Max. 300 mg						
**Darcie S, 2004** [41]		50 mg two times per day									
**Paran E, 1995** [42]									5–15 mg	40–120 mg	
**Munshi UK, 1992 ^2^** [43]					X						
**Bott-Kanner G, 1992** [44]									5–10 mg two or three times per day		
**Pickles CJ, 1989** [45]					100–200 mg three times per day						
**Ramanathan J, 1988 ^7^** [28]					X						
**Ashe RG, 1987** [46]					200 mg						
**Mabie WC, 1987** [47]					20–80 mg						
**Boutroy MJ, 1986** [48]	200–1200 mg per day										
**Macpherson M, 1986** [49]					100–300 mg three times per day						
**Högstedt S, 1985** [50]						50 mg two times per day (200 mg maximum per day)					
**Reynolds B, 1984** [51]		Max. 200 mg per day									
**Williams ER, 1983** [52]	300–600 mg per day										
**Rubin PC, 1983** [53]		100 mg per day									
**Liedholm H, 1983** [54]		50–200 mg per day				X					
**Liedholm H, 1983 ^2^** [27]		X									
**Livingstone I, 1983** [55]										30–160 mg per day	
**Dubois D, 1983** [26]	200 mg	100 mg							5 mg		
**Boutroy MJ, 1982** [56]	200–800 mg per day										
**Rubin PC, 1982** [57]		100 or 200 mg per day									
**Sandström B, 1982** [58]						100–400 mg					
**Garden A, 1982** [59]					200 mg						
**Dumez Y, 1981** [60]	200–800 mg/day										
**Bott-Kanner G, 1980** [61]										30–240 mg per day	
**O’Hare MF, 1980** [62]											200 mg per day
**Gallery ED, 1979 ^8^** [63]								X			
**Pruyn SC, 1979** [64]										10–80 mg per day	
**Eliahou HE, 1978** [65]										40–120 mg per day	

X = type of beta-blocker used in this study (in case the exact dosage is not provided). ^1^ This article did not mention the type of beta blocker that was used. ^2^ This article did not mention anything on the dose of the beta-blockers used. ^3^ A 20 mg intravenous bolus dose followed by 40 mg if not effective within 10 min, then 80 mg every 10 min until BP was lower than 150/100 mm Hg or until a maximum total dose of 220 mg was reached [36]. ^4^ A 20 mg bolus intravenous followed by a continuous infusion of 20 mg/h. When not effective within 20 min, this was followed by a 40 mg bolus and the continuous infusion was increased to 40 mg/h. When still not effective within 20 min an extra 80 mg bolus was followed by a continuous infusion that was increased up to 80 mg/h. Maximal cumulative dose was limited to 220 mg/h [37]. ^5^ Davis RL et al. [39] did not mention the types of beta-blockers which they included in their study. ^6^ A 20 mg intravenous bolus dose was followed by 40 mg if not effective within 20 min, followed by 80 mg every 20 min up to a maximum dose of 300 mg (five doses) [40]. ^7^ Before induction of anesthesia, 20 mg of labetalol was administered intravenously as a bolus followed by 10 mg increments every 2 min until the diastolic blood pressure was below 100 mmHg or the mean arterial pressure fell by 20% from baseline values [28]. ^8^ The dosage of the drug was altered as clinically indicated to maintain a sitting diastolic BP at or below 80 mm Hg [63].

**Table 4 ijerph-19-09616-t004:** Risk of bias in 18 randomized controlled trials.

Source	Domain 1. Risk of Bias from the Randomization Process	Domain 2. Risk of Bias Due to Deviations from the Intended Interventions	Domain 3. Missing Outcome Data	Domain 4. Risk of Bias in Measurement of the Outcome	Domain 5. Risk of Bias in selection of the Reported Result	Overall Risk of Bias Judgement
**Ashe RG, 1987** [46]	Some concerns	Low	Low	Low	Some concerns	High risk
**Bigelow CA, 2021** [29]	Low	Low	Low	Low	Low	Low
**Bott-Kanner G, 1992** [44]	Some concerns	Low	Low	Low	Some concerns	High risk
**Darcie S, 2004** [41]	Some concerns	Some concerns	Low	Low	Some concerns	High risk
**Easterling T, 2019** [33]	Low	Some concerns	Low	Low	Low	Some concerns
**Gallery ED, 1979** [63]	Some concerns	Some concerns	Low	Low	Some concerns	High risk
**Garden A, 1982** [59]	Some concerns	Some concerns	Low	Low	Some concerns	High risk
**Högstedt S, 1985** [50]	Some concerns	Some concerns	Low	Low	Some concerns	High risk
**Livingstone I, 1983** [55]	Some concerns	Some concerns	Low	Low	Some concerns	High risk
**Mabie WC, 1987** [47]	Some concerns	Some concerns	Low	Low	Some concerns	High risk
**Paran E, 1994** [42]	Some concerns	Some concerns	Low	Low	Some concerns	High risk
**Pickles CJ, 1989** [45]	Some concerns	Low	Low	Low	Some concerns	High risk
**Ramanathan J, 1988** [28]	Some concerns	Some concerns	Low	Low	Some concerns	High risk
**Reynolds B, 1984** [51]	Some concerns	Low	Low	Low	Some concerns	High risk
**Rubin PC, 1983** [53]	Some concerns	Low	Low	Low	Some concerns	High risk
**Singh R, 2016** [36]	Some concerns	Some concerns	Low	Low	Some concerns	High risk
**Verma R, 2012** [38]	Low	Some concerns	Low	Low	Some concerns	High risk
**Vigil-De Graca P, 2006** [40]	Low	Low	Low	Low	Some concerns	Some concerns

**Table 5 ijerph-19-09616-t005:** Risk of bias in 13 non-randomized controlled trials.

Source	Bias Due to Confounding	Bias in Selecting Participants	Bias in Classification of the Interventions	Bias Due to Deviations from Intended Interventions	Bias Due To Missing Data	Bias in Measurement of Outcomes	Bias in Selection of the Reported Result	Overall Risk of Bias
**Bateman BT, 2016** [35]	Moderate	Moderate	Moderate	Low	Low	Low	Moderate	Moderate
**Davis RL, 2011** [39]	Critical	Serious	Moderate	Low	NI	Low	Moderate	Critical
**Dumez Y, 1981** [60]	Critical	NI	Low	Low	Low	Low	Moderate	Critical
**Heida KY, 2012** [37]	NI	Low	Moderate	Low	Low	Low	Moderate	Moderate
**Kayser A, 2020** [30]	Moderate	Moderate	Moderate	Low	Low	Low	Low	Moderate
**Kumar N, 2020** [31]	Serious	Moderate	Moderate	Low	Low	Low	Moderate	Serious
**Liedholm H, 1983** [54]	Critical	Moderate	Serious	Low	Low	Low	Moderate	Critical
**Macpherson M, 1986** [49]	Serious	Serious	Low	Low	Low	Low	Moderate	Serious
**Mazkereth R, 2019** [32]	Critical	Serious	Serious	Low	Low	Low	Moderate	Critical
**Munshi UK, 1992** [43]	Moderate	Moderate	Serious	Low	Low	Low	Moderate	Serious
**Sandstrom B, 1982** [58]	Serious	Moderate	NI	Low	Low	Low	Moderate	Serious
**Thewissen L, 2017** [34]	Serious	Serious	Serious	Low	Low	Low	Moderate	Serious
**Williams ER, 1983** [52]	Serious	Moderate	Low	Low	Low	Low	Low	Serious

NI = No information.

**Table 6 ijerph-19-09616-t006:** Quality of 9 case series.

Source	Q1	Q2	Q3	Q4	Q5	Q6	Q7	Q8	Q9	Q10
**Boutroy MJ, 1986** [48]	Y	Y	Y	N	N	N	N	Y	N	N/A
**Liedholm, 1983** [27]	Y	Y	Y	Y	Y	N	N	Y	N	N/A
**Dubois, 1983** [26]	Y	Y	Y	Y	U	N	Y	Y	N	N/A
**Boutroy MJ, 1982** [56]	Y	Y	Y	N	N	N	N	Y	N	Y
**Rubin PC, 1982** [57]	Y	Y	Y	U	U	N	N	Y	N	Y
**Bott-Kanner G, 1980** [61]	N	Y	Y	N	N	N	Y	Y	N	N/A
**O’hare MF, 1980** [62]	Y	Y	Y	N	N	Y	Y	Y	N	Y
**Pruyn SC, 1979** [64]	N	Y	N	Y	Y	N	Y	Y	N	Y
**Eliahou HE, 1978** [65]	Y	Y	Y	Y	Y	N	Y	Y	N	N/A

Y = Yes, N= No, U = Unclear, N/A not applicable. Q1 Were there clear criteria for inclusion in the case series? Q2 Was the condition measured in a standard, reliable way for all participants included in the case series? Q3 Were valid methods used for identification of the condition for all participants included in the case series? Q4 Did the case series have consecutive inclusion of participants? Q5 Did the case series have complete inclusion of participants? Q6 Was there clear reporting of the demographics of the participants in the study? Q7 Was there clear reporting of clinical information of the participants? Q8 Were the outcomes or follow up results of cases clearly reported? Q9 Was there clear reporting of the presenting site(s)/clinic(s) demographic information? Q10 Was statistical analysis appropriate?

**Table 7 ijerph-19-09616-t007:** Quality of 15 case reports.

Source	Q1	Q2	Q3	Q4	Q5	Q6	Q7	Q8
**Sullo MG, 2015** [66]	N/A ^1^	N/A ^2^	Y	Y	Y	Y	Y	Y
**Stevens TP, 1995** [67]	N	N	Y	Y	Y	Y	Y	Y
**Klarr JM, 1994** [68]	Y	N	Y	U	Y	Y	Y	Y
**Haraldsson A, 1989** [69]	N/A ^1^	N/A ^2^	Y	Y	Y	Y	Y	Y
**Haraldsson A, 1989** [70]	N/A ^1,3^	N/A ^2^	Y	Y	U	Y	Y	Y
**Schmimmel MS, 1989** [71]	N/A ^1^	N/A ^2^	Y	Y	Y	Y	Y	Y
**Fox RE, 1985** [72]	Y	N	Y	Y	Y	Y	Y	Y
**Woods DL, 1982** [73]	N/A ^1^	N/A ^2^	Y	Y	Y	Y	Y	Y
**Bott-Kanner G,** **1978** [74]	U	Y	Y	Y	Y	U	Y	Y
**Sabom MB, 1978** [75]	N	N	Y	Y	Y	Y	Y	Y
**Datta S, 1978** [76]	Y	Y	Y	Y	Y	Y	Y	Y
**Habib A, 1977** [77]	N	N	Y	Y	Y	Y	Y	Y
**Cottrill CM,** **1977** [78]	N	N	U	Y	Y	Y	Y	Y
**Gladstone GR, 1975** [79]	N/A ^1,3^	N/A ^2^	Y	Y	Y	Y	Y	Y
**Fiddler GI, 1974** [80]	N	N	Y	Y	Y	Y	Y	Y

Abbreviations: Y = Yes; N= No; U = Unclear; N/A not applicable. Q1 Were patient’s demographic characteristics clearly described? Q2 Was the patient’s history clearly described and presented as a timeline? Q3 Was the current clinical condition of the patient on presentation clearly described? Q4 Were diagnostic tests or assessment methods and the results clearly described? Q5 Was the intervention(s) or treatment procedure(s) clearly described? Q6 Was the post-intervention clinical condition clearly described? Q7 Were adverse events (harms) or unanticipated events identified and described? Q8 Does the case report provide takeaway lessons? ^1^ This case report concerned a neonate in the first days of life. There was no information available about the neonates’ medical history, previous treatment and past diagnostic test results. ^2^ This case report concerned a neonate in the first days of life. There was no information available about the patients’ medical, family and psychosocial history (including relevant information, as well as relevant past interventions and their outcomes). ^3^ Diagnosis, treatment/medication and medical history of the mother were described, but not of the neonate.

**Table 9 ijerph-19-09616-t009:** Beta-blockers compared to other antihypertensive drugs or placebo in mothers with gestational hypertension.

Certainty Assessment							Summary of Findings				
Participants (Studies)Follow-up	Risk of Bias	Inconsistency	Indirectness	Imprecision	Publication Bias	Overall Certainty of Evidence	Study Event Rates (%)		Relative Effect (95% CI)	Anticipated Absolute Effects	
							With Other Antihypertensive Drugs or Placebo	With Beta-Blockers		Risk with Other Antihypertensive Drugs or Placebo	Risk Difference with Beta-Blockers
**Bradycardia (Cohort studies)**											
2,293,407 (3 observational studies)	Serious ^a^	Serious ^b^	not serious	Serious ^c^	strong association	⨁⨁◯◯ Low	11,669/2,282,318 (0.5%)	175/11089 (1.6%)	**RR 2.10** (0.97 to 4.55)	5 per 1000	**6 more per 1000** (from 0 fewer to 18 more)
**Bradycardia (Case control studies)**											
45 cases 800 controls (5 observational studies)	very serious ^d^	very serious ^e^	not serious	very serious ^f^	none	⨁◯◯◯ Very low	45 cases 800 controls		**RR 1.86** (0.31 to 11.19)	Low	
										0 per 1000	**0 fewer per 1000** (from 0 fewer to 0 fewer)
**Bradycardia (RCTs)**											
2013 (11 RCTs)	very serious ^g^	not serious	not serious	not serious	none	⨁⨁◯◯ Low	15/1017 (1.5%)	38/996 (3.8%)	**RR 2.36** (1.33 to 4.18)	15 per 1000	**20 more per 1000** (from 5 more to 47 more)
**Hypoglycemia (Cohort studies)**											
2,374,957 (6 observational studies)	Serious ^h^	serious ^i^	not serious	not serious	strong association	⨁⨁⨁◯ Moderate	29,954/2,362,830 (1.3%)	620/12,127 (5.1%)	**RR 3.01** (2.79 to 3.25)	13 per 1000	**25 more per 1000** (from 23 more to 29 more)
**Hypoglycemia (Case control studies)**											
168 cases 416 controls (5 observational studies)	very serious ^j^	not serious ^i^	not serious	Serious ^k^	none	⨁◯◯◯ Very low	168 cases 416 controls		**RR 1.72** (1.33 to 2.22)	Low	
										0 per 1000	**0 fewer per 1000** (from 0 fewer to 0 fewer)
**Hypoglycemia (RCTs)**											
2051 (12 RCTs)	Serious ^l^	not serious	not serious	serious ^c^	none	⨁⨁◯◯ Low	57/1018 (5.6%)	77/1033 (7.5%)	**RR 1.06** (0.79 to 1.41)	56 per 1000	**3 more per 1000** (from 12 fewer to 23 more)

**CI:** confidence interval; **RR:** risk ratio. **Explanations** ^a^ All the studies had a moderate risk of overall bias. ^b^ I2 > 50%. ^c^ 95% CI crosses line of no effect. ^d^ The study with the highest weightage had a serious risk of overall bias. ^e^ I2 = 82%. ^f^ Very low event rates and 95% CI shows appreciable benefit and harm. ^g^ Most of the studies had a high risk of overall bias. ^h^ The study contributing to maximum weightage had a moderate risk of overall bias. ^i^ Though the I2 was large, it was due to differences between the small and large magnitude of effects. ^j^ Most of the studies had a serious to critical risk of overall bias. ^k^ Optimal information criterion (OIS) not satisfied due to a low event rate and sample size. ^l^ The studies with highest weightage had a high risk of overall bias or some concerns. ⨁◯◯◯: Very low, ⨁⨁◯◯: Low, ⨁⨁⨁◯: Moderate and ⨁⨁⨁⨁: High.

**Table 10 ijerph-19-09616-t010:** Studies with hypoglycemia as outcome measure.

Article	Study Groups	Hypoglycemia	Blood Glucose (mmol/L)	Definition of Hypoglycemia	Time of Control of Hypoglycemia
**Bigelow CA, 2021** [29]	Propranolol vs. placebo	11/45 (24.4%) vs. 8/49 (16.3%) (*p* = 0.33)		Blood sugar < 40 md/dL	N
**Kayser A, 2020** [30]	ß-blocker vs. methyldopa ß-blocker vs. non-hypertensive mother	7/252 (2.8%) vs. 3/199 (1.5%) (NS) 7/252 (2.8%) vs. 13/588 (2.2%) (NS)		Blood glucose < 35 mg/dl at the first day of life or <45 mg/dL after the first day of life	N
**Kumar N, 2020** [31]	ß-blocker vs. no disease	78/228 (34.6%) vs. 914/4103 (22.2%) (*p* < 0.01)		Blood glucose < 40 mg/dL	At least 30 min after feeding. Feeding was initiated as soon as possible after delivery. For at least 24 h in late preterm and small for gestational age (SGA), and for the first 12 h in LGA infants and infants of mothers with diabetes
**Mazkereth R, 2019** [32]	ß-blocker vs. control	47/153 (30.7%) vs. 28/153 (18.3%) *p* = 0.016		Glucose < 40 mg/dL on the first day of life	Hours 1, 2, 4 and 6 of life and every 8 h thereafter (to complete a 48-h follow-up)
**Easterling T, 2019** [33]	Labetalol vs. nifedipine Labetalol vs. methyldopa	0/290 (0%) vs. 1/298 (<1%) 0/290 (0%) vs. 0/294 (0%)		N	N
**Bateman BT, 2016** [35]	ß-blocker vs. control Labetalol vs. control Metoprolol vs. control Atenolol vs. control Propensity-score ^1^ corrected available	460/10,585 (4.3%) vs. 27,228/2,281,531 (1.2%) 345/6748 (5.1%) vs. 27,228/2,281,531 (1.2%) 49/1485 (3.3%) vs. 27,228/2,281,531 (1.2%) 30/1121 (2.7%) vs. 27,228/2,281,531 (1.2%)		Glucose ≤ 35 mg/dL	N
**Singh R, 2016** [36]	Labetalol vs. hydralazine	NS		N	N
**Heida KY,** **2012** [37]	Labetalol vs. control Labetalol i.v. vs. labetalol oral	26/55 (47.3%) vs. 23/54 (42.6%) *p* = 0.62 43.2 vs. 55.6% *p* = 0.45		Glucose < 2.7 mmol/L	In the first 48 postnatal hours
**Verma R, 2012** [38]	Labetalol vs. methyldopa	1/45 (2.22%) vs. 2/45 (4.44%) (NS)		N	N
**Davis RL, 2011 ^2^** [39]	ß-blocker vs. control ß-blocker vs. calcium channel blockers	34/405 (8.4%) vs. 1771/75,688 (2.3%) 34/405 (8.4%) vs. 25/721 (3.5%)		N	N
**Vigil-De Gracia P, 2006** [40]	Labetalol vs. hydralazine	6/103 (5.8%) vs. 6/102 (5.8%)		Plasma glucose < 35 mg/dL	N
**Darcie S, 2004** [41]	Atenolol vs. isradipine Atenolol vs. control	26/40 (65%) vs. 24/39 (61.5%) (*p* = 0.818) 26/40 (65%) vs. 4/14 (28.5%) (*p* < 0.05)		Blood glycemia levels < 40 mg/dL	1, 3, 6, 12 and 24 h after birth
**Paran E, 1995** [42]	Hydralazine vs. hydralazine and propranolol Hydralazine vs. hydralazine and pindolol		76.4 ± 16.5 vs. 62.6 ± 14.0 mg% (*p* < 0.02) 76.4 ± 16.5 vs. 78.6 ± 15.7 mg% (*p* < 0.02)	N	In the first 48 postnatal hours
**Munshi UK, 1992** [43]	Labetalol vs. control	23/48 (47.9%) vs. 14/81 (17.2%) (*p* < 0.01)		Blood glucose value of <30 mg/dL irrespective of gestational age, within the first 72 h of life and below 40 mg/dL thereafter	First at 1–2 h of age and again at 4–6 h of age, thereafter 2–6 hourly depending on the previous blood glucose results. The monitoring was stopped once at least two blood glucose values were above 40 mg on an oral feeding alone
**Bott-Kanner G, 1992** [44]	Pindolol vs. placebo	^3^		Blood glucose < 25 mg/dL	During the first 24 h of life
**Pickles CJ, 1989** [45]	Labetalol vs. placebo	4/70 (5.7%) vs. 3/74 (4.1%)		Blood glucose < 1.4 nmol/L	N
**Ramanathan J, 1988** [28]	Labetalol vs. control	0/15 (0%) vs. 0/10 (0%)	53.4 ± 2.8 vs. 50 ± 3.1 (NS)	N	Within 10 to 20 min of delivery
**Ashe RG, 1987** [46]	Labetalol vs. hydralazine	1/10 (10%) vs. 0/10 (0%)		N	Every 4 h, for 1 day
**Mabie WC, 1987** [47]	Labetalol vs. hydralazine	1/13 (7.7%) vs. 1/6 (16.7%)		Blood glucose < 35 mg/dL	N
**Macpherson M, 1986** [49]	Labetalol vs. control	^4^		<35 mg/dL	At 2, 4, 8,16, 24, 48 and 72 h after birth
**Högstedt S, 1985** [50]	Metoprolol and hydralazine vs. control (intended-to-treat) ^5^ Metoprolol and hydralazine vs. control (cause-effect)	9/82 (11.0%) vs. 11/79 (13.9%) 8/69 (11.6%) vs. 10/66 (15.2%)		Blood glucose ≤ 1.7 mmol/L	N
**Williams ER, 1983** [52]	Acebutolol vs. methyldopa	0/9 vs. 0/11		N	N
**Rubin PC, 1983** [53]	Atenolol vs. placebo	1/46 (2.2%) vs. 4/39 (10.3%) (NS)		Confirmed serum glucose < 1.4 nmol/L	At 1, 4, 6, 12 and 24 h
**Liedholm H, 1983** [54]	Atenolol or metoprolol (no control group)	4/95 (4.2%)		N	N
**Livingstone I, 1983** [55]	Propranolol vs. methyldopa	2/14 (14.3%) vs. 0/14 (0%)		N	For 48 h after delivery
**Dubois D, 1983** [26]	Beta blocker (no control group)	1/125 (0.8%)		N	At birth
**Rubin PC, 1982** [57]	Atenolol (no control group)	2/9 (22%)		N	During the first 24 h of life
**Sanström B, 1982** [58]	Bendroflumethiazide + metoprolol vs. metoprolol + hydralazine vs. Bendroflumethiazide + hydralazine	^6^		N	N
**Dumez Y, 1981** [60]	Acebutolol vs. methyldopa	6/10 vs. 1/10	Day 1: 1.60 mmol/L ± 0.99 vs. 2.55 mmol/L ± 0.42 (NS), Day 2: 2.63 mmol/L ± 0.50 vs. 2.47 mmol/L ± 0.63 (NS), Day 3: 3.29 mmol/L ± 1.53 vs. 2.72 mmol/L ± 1.19 (NS)	N	Daily within the 3 first days of life, for the first time at about three hours of life and on the second and third days of life two hours after feeding
**Bott-Kanner G, 1980** [61]	Propranolol (no control group)	2/14 (14.3%)		Blood glucose < 35 mg/dL	Frequently In the first few hours of life
**O’Hare MF, 1980** [62]	Sotalol (no control group)	1/12 (8.33%)		N	Four-hourly
**Gallery ED, 1979** [63]	Oxprenolol vs. methyldopa		3.8 ± 0.27 vs. 2.8 ± 0.36 mmol/L *p* < 0.05	N	N
**Pruyn SC, 1979** [64]	Propranolol (no control group)	3/12 (25%)		N	N
**Eliahou HE, 1978** [65]	Propranolol (no control group)	0/22 (0%)		N	N

N: is not described in article. NS: not significant. ^1^ PS-matched: Propensity scores were estimated using a logistic regression model in which exposure was the dependent variable and was estimated on the basis of 5 groups of potential confounders of the planned analysis: demographic characteristics, medical conditions, obstetrical conditions, maternal medications, and measures of healthcare use [35]. ^2^ Hypoglycemia was grouped under endocrine and metabolic disturbances specific to newborns, which included neonatal hypoglycemia [39]. ^3^ As regards other outcome variables, namely, the Apgar score, respiratory and heart rate at delivery, hypoglycemia and jaundice during the first 24 h—the differences between the two treatment groups were inconsistent and non-significant [44]. ^4^ Blood glucose levels were mostly within the normal reported range of 2.0–5.0 mmol/L for term infants, but they all tended towards the lower limit, with a range of 1.8–4.2 mmol/L, with a median of 3.2 mmol/l. No infant was clinically hypoglycemic at any time [49]. ^5^ For the analyses, the material was divided into two categories. The first group gives data for all the 161 patients for whom it was the intention to treat. In the calculation of cause-and-effect, 26 patients were withdrawn from the original group of 161: in 5 patients of C-group, DBP exceeded 110 mmHg and they were then treated with antihypertensive drugs; one patient in the T-group admitted that she had not taken the prescribed drugs; 6 patients gave birth to malformed or stillborn children and their data were not used for the calculation of Apgar scores, birth weights or other vitality signs. Eight patients in the T-group and 6 in the C-group gave birth within 2 weeks after admission to the study, and these 14 women were excluded from the cause-and-effect analyses because of the short treatment period [50]. ^6^ There were no abnormal changes in heart frequency, P-glucose, P-bilirubin and maturity of the lungs of the new-born infants in groups A and B in comparison with group C, in which no adrenergic beta-blocking agent was used [58].

**Table 11 ijerph-19-09616-t011:** Overview case-reports.

Source	Country	Treatment Indication	Beta-Blocker Type	Dosage	Blood Glucose	Heart Rate (Beats per Minute)
**Sullo MG, 2015** [66]	Italy	Unspecified tachycardia	Nebivolol	5 mg per day	30 mg/dL	x
**Stevens TP, 1995** [67]	United States	Hypertension	Labetalol	150 mg two times per day	1.7 mmol/L (30 mg/dL)	111 after birth 100 in transit to tertiary referral center 100–120 (resting heart rate)
**Klarr JM, 1994** [68]	United States	Hypertension prior to cesarean section	Labetalol	Single 30 mg dosage	31 mg/dL (twin A) 37 mg/dL (twin B)	<80 (both twins)
**Haraldsson A, 1989** [69]	The Netherlands	Hypertension prior to cesarean section	Labetalol	50 mg/hour	Not described in article, but intravenous glucose was given	Severe bradycardia immediately after delivery. On admission the heart rate was 140
**Haraldsson A, 1989** [70]	The Netherlands	Pregnancy induced hypertension	Labetalol	600 mg (200 mg 3 times per day)	1.2 mmol/l	<80 after delivery, 148 later
**Schmimmel MS, 1989** [71]	Israel	Postpartum hypertension	Atenolol	100 mg daily (50 mg two times per day)	x	80
**Fox RE, 1985** [72]	United States	Hypertension	Nadolol	20 mg once per day	20 mg/dl	136 after birth 112 at 4.5 h of age ^1^ >135 at 4 days of age
**Woods DL, 1982** [73]	South Africa	Uncontrolled hypertension	Atenolol	100 mg daily	2–5 mmol/L (45 mg/100 mL)	138
**Bott-Kanner G,****1978** [74]	Israel	Chronic hypertension	Propranolol	160 --> 60 mg/day ^2^ (first pregnancy) 120 --> stop (second pregnancy)	37 mg/dL (first pregnancy) 75 mg/dL after birth, 87 mg/dL next day (second pregnancy)	150 (first pregnancy) 140 (second pregnancy)
**Sabom MB, 1978** [75]	United States	Idiopathic Hypertrophic subaortic stenosis	Propranolol	60 mg 4 times per day, discontinued upon admission	25–45 mg/100 ml	Sinus bradycardia (heart rate not mentioned)
**Datta S, 1978** [76]	United States	Idiopathic Hypertrophic subaortic stenosis	Propranolol	80 mg daily	No hypoglycemia	110–120, with a short period of 80 during sleep
**Habib A, 1977** [77]	United States	Hyperthyroidism and congestive heart failure (case 1), supraventricular tachycardia (case 2 and 3) and hyperthyroidism (case 4)	Propranolol	10 mg 4 times per day (case 1, 2 and 3), 10 mg two times per day (case 4) ^3^	20 mg/dL (case 1) 30 mg/dL (case 2) 25 mg/dL at one hour of age and 5 mg/dL at two hours of age (case 3) 25–45 mg/dL (case 4)	80 within one hour of delivery (case 1) ^4^ 80–90 ^5^ (case 2) 100–120 ^6^ (case 3) 80–90, frequently during first 24 h of life (case 4)
**Cottrill CM,** **1977** [78]	United States	Chronic atrial tachycardia	Propranolol	160 mg per day (40 mg 4 times per day) ^7^	Too low to be detected by the Dextrostix method	40 at birth, later it varied between 100 and 165
**Gladstone GR, 1975** [79]	United States	Essential hypertension	Propranolol	240 mg per Day decreased to 160 mg per Day ^8^	11 mg/dL	70–90 during first day of life. Rose with stimulation to 120/minute. Was 120–130/minute on day 5
**Fiddler GI, 1974** [80]	Scotland	Hypertrophic Obstructive cardiomyopathy	Propranolol	30 mg 3 times per day	12 mg/100 mL	x

^1^ The heart rate remained low for 72 h, and frequent short episodes of bradycardia occurred that were not associated with apnea and that resolved spontaneously [72]. ^2^ At the time of conception, the mother was placed on a regimen of 160 mg/day. Three weeks prior to term, the propranolol dosage was reduced to 60 mg/day in preparation for the delivery [74]. ^3^ In case 1, the mother was receiving propranolol 20 mg for times a day at time of delivery [77]. ^4^ The bradycardia persisted for most of the first 36 h of life, and the heart rate remained between 100 to 120 per minute for most of the second 36 h [77]. ^5^ Frequent episodes of bradycardia during the first 24 h of life and occasional episodes over the next 48 h [77]. ^6^ The infant had occasional episodes of bradycardia (heart rate 80 to 90/minute) during the first 48 h of life, but his heart rate generally ranged between 100 to 120/minute [77]. ^7^ The day prior to the Cesarean section, she received 60 mg propranolol every six hours and 60 mg was given orally six hours before surgery; an additional 3 mg was given intravenously one hour before the operation [78]. ^8^ The mother was taking 240 mg/day at the time of conception. The dose was decreased to 160 mg/day in the fourth month of pregnancy [79].

## Data Availability

Not applicable.

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
