# Peer review of "The Risk for Neonatal Hypoglycemia and Bradycardia after Beta-Blocker Use during Pregnancy or Lactation: A Systematic Review and Meta-Analysis"

_ijerph, 2022, doi:10.3390/ijerph19159616_

Round 1

Reviewer 1 Report

The study is very interesting. The funnel plot should be added and the impact of the half-life, lipophilicity of the product, the average dose used and the volume of distribution should be discussed.

Author Response

Reviewer 1

The study is very interesting. The funnel plot should be added and the impact of the half-life, lipophilicity of the product, the average dose used and the volume of distribution should be discussed.

Response: We thank the reviewer for the comment and the suggestions.

We have included funnel plots along with the Egger’s test and Regression test for the randomized controlled trials and for the proportion-based meta-analysis respectively in our revised manuscript (Figure 2 a, b and Figure 7 a, b). The funnel plots of the included RCTs indicated no publication bias for both hypoglycemia and bradycardia, while we did detect publication bias for the proportion based meta-analyses for the aforementioned outcomes. We would like to indicate that publication bias could be assessed only if we have sufficient number of studies assessing an intervention versus comparator (at least 10) and henceforth, it could not be assessed for many of the other meta-analyses.

These are now described in the revised manuscript:

Page 25: “Funnel plot evaluation and Egger’s test of the included RCTs in the meta-analyses for the outcome bradycardia indicated no publication bias (Figure 2 a). The regression test however indicated publication bias for the proportion based meta-analysis for bradycardia (Figure 2 b)”

Page 39: “While the funnel plot of the included RCTs and the Egger’s test indicated no publication bias, we did detect publication bias for the proportion based meta-analysis based on the regression test (Figure 7 a, b).”

Moreover, we have added the following sentence to the discussion on page 44 of the manuscript: “Fetal and neonatal drug exposure and effects following maternal pharmacotherapy are influenced by multiple drug specific characteristics, i.a. lipophilicity, neonatal half-life, different dosages, the duration of maternal exposure as well as via lactation, the aspects of placental passage of beta-blockers, intra- and inter-patient variability in disposition, and their varying potency on neonatal bradycardia and hypoglycemia. Generally, fetal and neonatal exposure and effects are increased for drugs with longer neonatal elimination half-life, the more lipophilic drugs (leading to increased placental passage as well as disposition to breastmilk), increased dosages, and increased duration of maternal treatment. This profile has not yet been reported for each beta-blocker used in clinical practice and requires further investigation.”

Reviewer 2 Report

The work i excellent and deals with important issue of the effect of the drugs used (beta-blockers) in the mother on the condition of the newborn. The English is flawless. Nevertheless, there were a few minor shortcomings in the work that do not detract from its value:
It did not distinguish between cases of one-time use of beta blockers (induction of labor, induction of anesthesia for cesarean section) and long-term use of beta blockers due to maternal illness. For this reason, I would suggest excluding the work of Bigelow et al. (2021) and Ramanathan et al. (1998) from the analysis - Table 2 and 10, as well as that of Klarr et al. (1994) - and analyzing cases of long-term use of beta-blockers only.
It would be helpful to clarify in Table 3 what the letter X means and in Table 5 what NI means.
In Table 6, column Q9 is redundant and in column Q10, instead of NA, it should be N/A to be consistent with the abbreviations.
In Table 7 - the article by Gladstone et al. (1975) should be excluded because it does not contain the medical history of the newborn.
The papers of Easterling et al. (2019a and b), on the other hand, should be excluded from the analysis (Figures 2 and 6) because they do not include the use of beta-blockers. Also so the work of Williams et al. (1983) (Figures 3,5 and 6) should not be considered, as it does not affect the results.
Having taken my suggestions into consideration, I strongly support the acceptance of the paper for publication.

Author Response

Reviewer 2

The work is excellent and deals with important issue of the effect of the drugs used (beta-blockers) in the mother on the condition of the newborn. The English is flawless. Nevertheless, there were a few minor shortcomings in the work that do not detract from its value.

Response: We thank the reviewer for the very positive remarks. The suggestions for improvements were very valuable and we have incorporated them in our revised manuscript.

It did not distinguish between cases of one-time use of beta blockers (induction of labor, induction of anesthesia for cesarean section) and long-term use of beta blockers due to maternal illness. For this reason, I would suggest excluding the work of Bigelow et al. (2021) and Ramanathan et al. (1998) from the analysis - Table 2 and 10, as well as that of Klarr et al. (1994) - and analyzing cases of long-term use of beta-blockers only.

Response: We thank the reviewer for pointing this out. We agree with the view of the reviewer and have excluded the studies of Bigelow et al. and Ramanathan et al. from the meta-analyses. We are of the view that it would be appropriate to present these studies as a narrative in our systematic review since we aimed to give a comprehensive overview of the neonatal effects after all types of maternal beta-blocker use.

We have updated Figure 1 (Overview literature search) and all figures and tables regarding the meta-analyses. Moreover, we have added the following sentences to the manuscript to specify that the studies of Bigelow et al. and Ramanathan et al. focused on short term use of beta-blockers and are excluded from the meta-analysis:

  • Page 5: “Furthermore, two other studies were excluded from the meta-analysis since these focused on short term use of beta-blockers during pregnancy for induction of anesthesia or labor in-stead of the use for cardiovascular diseases [38, 45].”
  • Page 21: ”In one of these studies the mother received Labetalol solely as a short-term pretreatment for anesthesia prior to cesarean section [38].”
  • Page 37: “The study of Ramanathan et al. [38] did not find an increased risk for hypoglycemia. However, in this study labetalol was prescribed for short term use solely for induction of anesthesia.” and “However, in the study of Bigelow et al. [45] mothers received propranolol solely short term for induction of labor.”

The study of Klarr et al. was not included in the meta-analyses because it is a case-report.

It would be helpful to clarify in Table 3 what the letter X means and in Table 5 what NI means.

Response: We have added the following information to the legend of Table 3: “X = type of beta-blocker used in this study (in case the exact dosage is not provided) ” and “NI = No information” to the legend of Table 5. 

In Table 6, column Q9 is redundant and in column Q10, instead of NA, it should be N/A to be consistent with the abbreviations.

Response: We have now changed NA into N/A. Regarding Q9 we believe it is important to present all data even though it is not available for all case series.

In Table 7 - the article by Gladstone et al. (1975) should be excluded because it does not contain the medical history of the newborn.

Response: On forehand we decided to include all English written studies reporting the adverse effects of beta-blocker exposure during pregnancy and lactation on the neonate in human subjects. Our exclusion criteria did not specify that studies would be excluded solely based on the fact that the information provided by the study is limited. Therefore we would like to present the data of this case report. Since this study is a case report it was not included in the meta-analysis. 

The papers of Easterling et al. (2019a and b), on the other hand, should be excluded from the analysis (Figures 2 and 6) because they do not include the use of beta-blockers. Also so the work of Williams et al. (1983) (Figures 3,5 and 6) should not be considered, as it does not affect the results.

Response: The study of Easterling, T., et al. (Oral antihypertensive regimens (nifedipine retard, labetalol, and methyldopa) for management of severe hypertension in pregnancy: an open-label, randomised controlled trial. Lancet, 2019.) compared Labetalol and Nifedipine with Labetalol and Methyldopa. Therefore, we found it acceptable to include this study in our systematic review and meta-analyses.

The study of Williams et al. indeed had zero events in both the case group and the control group. However, in line with the Cochrane guidelines even studies with no event rates need to be included in the meta-analysis since they add to the total number of patients. These could affect the imprecision parameter included in the GRADE working group guidelines for certainty of evidence evaluation. 

Having taken my suggestions into consideration, I strongly support the acceptance of the paper for publication.

Response: We thank the reviewer for the positive remark.

Reviewer 3 Report

Thank you for giving me the opportunity for review the manuscript entitled  “The Risk for Neonatal Hypoglycemia and Bradycardia After Beta-Blocker use During Pregnancy or Lactation: A Systematic  Review and Meta-Analysis”. The manuscript is interesting and in scope of the Journal. The topic of the work is demanding but correctly presented. The collected data are presented in extensive tables, sometimes too long. Figures and tables are informative. The manuscript is interesting.  The systematic review was conducted according to accepted standards (PRISMA) - a great advantage. The manuscript also includes a meta-analysis which adds to the value of the work.

I rate the presented article very positively.

Author Response

Reviewer 3

Thank you for giving me the opportunity for review the manuscript entitled  “The Risk for Neonatal Hypoglycemia and Bradycardia After Beta-Blocker use During Pregnancy or Lactation: A Systematic  Review and Meta-Analysis”. The manuscript is interesting and in scope of the Journal. The topic of the work is demanding but correctly presented. The collected data are presented in extensive tables, sometimes too long. Figures and tables are informative. The manuscript is interesting.  The systematic review was conducted according to accepted standards (PRISMA) - a great advantage. The manuscript also includes a meta-analysis which adds to the value of the work.

I rate the presented article very positively.

Response: We thank the reviewer for the positive remarks, time and careful attention for the review of our manuscript. We agree that the tables are very long. Therefore, some of the tables are provided supplemental only files.